# Mitochondrially targeted tamoxifen alleviates markers of obesity and type 2 diabetes mellitus in mice

Eliska Vacurova [1,2,10], Jaroslava Trnovska[3,10], Petr Svoboda [3,4,10], Vojtech Skop [5], Vendula Novosadova[6], David Pajuelo Reguera [6], Silvia Petrezselyová [6], Benoit Piavaux[6], Berwini Endaya[1], Frantisek Spoutil[6], Dagmar Zudova[6], Jan Stursa[1], Magdalena Melcova [4], Zuzana Bielcikova [7], Lukas Werner[1], Jan Prochazka[6], Radislav Sedlacek [6], Martina Huttl[3], Sona Stemberkova Hubackova [1✉], Martin Haluzik [8✉] & Jiri Neuzil [1,9✉]

Type 2 diabetes mellitus represents a major health problem with increasing prevalence worldwide. Limited efficacy of current therapies has prompted a search for novel therapeutic options. Here we show that treatment of pre-diabetic mice with mitochondrially targeted tamoxifen, a potential anti-cancer agent with senolytic activity, improves glucose tolerance and reduces body weight with most pronounced reduction of visceral adipose tissue due to reduced food intake, suppressed adipogenesis and elimination of senescent cells. Glucose-lowering effect of mitochondrially targeted tamoxifen is linked to improvement of type 2 diabetes mellitus-related hormones profile and is accompanied by reduced lipid accumulation in liver. Lower senescent cell burden in various tissues, as well as its inhibitory effect on pre-adipocyte differentiation, results in lower level of circulating inflammatory mediators that typically enhance metabolic dysfunction. Targeting senescence with mitochodrially targeted tamoxifen thus represents an approach to the treatment of type 2 diabetes mellitus and its related comorbidities, promising a complex impact on senescence-related pathologies in aging population of patients with type 2 diabetes mellitus with potential translation into the clinic.

[1] Institute of Biotechnology, Czech Academy of Sciences, Prague-West, Czech Republic. [2] Faculty of Science, Charles University, Prague, Czech Republic. [3] Centre for Experimental Medicine, Institute for Clinical and Experimental Medicine, Prague, Czech Republic. [4] Department of Biochemistry and Microbiology, University of Chemistry and Technology Prague, Prague, Czech Republic. [5] Diabetes, Endocrinology, and Obesity Branch, National Institute of Diabetes and Digestive and Kidney Diseases, NIH, Bethesda, MD 20892, USA. [6] Institute of Molecular Genetics, Czech Academy of Sciences, Prague-West, Czech Republic. [7] General University Hospital, Prague, Czech Republic. [8] Diabetes Centre, Institute for Clinical and Experimental Medicine, Prague, Czech Republic. [9] School of Pharmacy and Medical Science, Griffith University, Southport, QLD, Australia. [10] These authors contributed equally: Eliska Vacurova, Jaroslava Trnovska, Petr Svoboda. ✉email: sona.stemberkova@ibt.cas.cz; halm@ikem.cz; j.neuzil@griffith.edu.au

The increasing prevalence of type 2 diabetes mellitus (T2DM) with its chronic debilitating complications represents one of the most significant health threats worldwide[1]. Diabetes, currently affecting 422 million people around the globe, is expected to become the seventh leading cause of death by 2030[1]. This is particularly evident in elderly patients, where it can affect as many as 30–40% of the population compared to ~6–25% of patients under 65 years of age[2].

A combination of genetic predisposition, sedentary lifestyle, and excessive intake of calorie-rich food leads to obesity that, in turn, considerably increases the risk of T2DM development[3,4]. Excessive accumulation of adipose tissue and its functional changes largely contribute to the etiopathogenesis of T2DM and accompanying diseases, such as arterial hypertension, dyslipidemia, non-alcoholic fatty liver disease (NAFLD), and many other pathologies[5]. In patients with obesity, markedly enlarged adipocytes, in particular in the visceral fat compartment, are more prone to apoptosis that stimulates mobilization of pro-inflammatory macrophages and other immunocompetent cells into visceral fat[6]. As a result, adipose tissue produces excessive amounts of pro-inflammatory factors that are released into the circulation and contribute to the development of sub-clinical inflammation, typically present in patients with T2DM and obesity[7]. Furthermore, chronic/enhanced ectopic lipid accumulation in non-adipose tissues contributes to the development of insulin resistance in the liver and muscles, and to increased apoptosis of insulin-producing β-cells in the pancreas[8].

Changes induced by long-standing, poorly controlled obesity followed by the development of T2DM are linked to premature senescence in various tissues, contributing to further deterioration of their function and eventually to the development of chronic irreversible complications[9]. Cellular senescence is a form of cell cycle arrest that limits the proliferative potential of cells[10]. Despite the arrest, senescent cells are metabolically active and produce various cytokines, chemokines, proteases, and growth factors (collectively referred to as "senescence-associated secretory phenotype", SASP) that affect the surrounding environment[11]. The inability of immune cells to eliminate senescent cells from the organism results in chronic inflammation and gradual tissue damage, as seen in the panoply of age-related pathologies, including T2DM[12].

Senescent cells play an important role in the induction of T2DM pathogenesis[13]. Considering that metabolic and signaling changes associated with T2DM can promote senescence, senescent cells are components of the "pathogenic loop" in diabetes. In obese and diabetic mice, visceral adipose tissue (VAT) is the most prominent compartment of senescent cells accumulation. VAT, therefore, presents the nexus of mechanisms involved in longevity and age-related metabolic dysfunctions[14]. A close relationship between visceral fat content and the risk of T2DM and cardio-vascular complications has also been demonstrated in humans. Components of SASP secreted by adipose-derived senescent cells confer insulin resistance to metabolic tissues and attract immune cells that can exacerbate the effects of insulin resistance[14,15]. Moreover, there is a close relationship between senescence and fat accumulation in hepatocytes followed by the development of steatosis in diabetic mice[16].

Senolytic agents may improve glucose control and obesity- and diabetes-related pathologies[14], supporting the idea that targeting senescent cells may be a promising strategy for T2DM management. Mitochondrial function is an important determinant of the aging process (reviewed in ref. [17]), and we have recently reported that targeting mitochondria in senescent cells presents a plausible way to eliminate such cells in the context of pathological senescence as well as senescence-associated diseases[18]. Using mitochondrially targeted tamoxifen (MitoTam), our proprietary agent

with anticancer activity[19] that has recently undergone Phase 1/1b clinical trial (EudraCT 2017-004441-25), we have achieved specific elimination of senescent cells. Treatment with MitoTam effectively reduces oxidative phosphorylation (OXPHOS) and mitochondrial membrane potential in senescent cells, and severely affects mitochondrial morphology based on a low level of the ADP/ATP translocation channel ANT2 (adenine nucleotide translocase 2). These cells cannot, therefore, pump ATP inside mitochondria in order to maintain mitochondrial potential by cleavage of ATP by ATPase, resulting in the collapse of mitochondrial integrity and function[18]. Based on these results, we reasoned that MitoTam may present a non-cannonical therapeutic modality to treat senescence-associated pathologies, such as T2DM.

Here we show that MitoTam considerably improves glucose control, decreases body weight, and reduces diabetic markers as well as diabetic comorbidities in mice with diet-induced obesity and prediabetes. These improvements are associated not only with a reduction of food intake (FI) and a drop in the number of senescent cells in the organism but also with rejuvenation of the adipose tissue, suggesting the role of MitoTam in T2DM treatment and prevention of chronic diabetic complications.

## Results

**MitoTam improves metabolic parameters in aged mice**. The process of aging is often accompanied by alternations in glucose metabolism and associated complications, including glucose intolerance, insulin resistance, and pancreatic β-cell dysfunction[13]. Since we have recently shown that the potential anticancer compound, MitoTam[19], also acts as an efficient senolytic agent[18], we have decided to test whether removing age-related senescent cells may improve the metabolic phenotype. Thus, naturally aged (18 months) and young (8 weeks) C57BL/6 mice were treated with MitoTam, and metabolic parameters, as well as the level of senescence markers in different organs, were assessed. As expected, all tested organs from aged animals were positive for senescence-associated β-galactosidase (SA-β-gal) and featured elevated levels of senescence markers $p21^{waf1}$, $p16^{Ink4a}$ or $p19^{Ink4d}$ (Fig. 1a–e, Supplementary Fig. 1a–c). This increase was reversed by MitoTam treatment, confirming its senolytic activity. Aged mice also showed increased fasting glucose (Fig. 1f) and plasma leptin levels (Fig. 1g), along with the increased weight of VAT (comprising the epididymal adipose tissue—EAT and perirenal adipose tissue—PRAT) (Supplementary Fig. 1e), suggesting higher adiposity and impaired metabolism. Treatment with MitoTam improved most of these parameters in aged mice and shifted them to levels similar to those in young animals without any considerable effect on body weight or FI (Supplementary Fig. 1d, f). These results indicate plausibility to study MitoTam in a relevant model of metabolic disorder.

Inflammation and accumulation of pro-inflammatory cytokines have a direct negative impact on pancreatic β-cell function, adiponectin suppression, and subsequent insulin resistance in obesity and T2DM[20–22]. Senescent cells, as important producers of these cytokines, play an essential role in the progression of the diseases. To corroborate this notion, we assessed the presence of senescent cells and senescence-associated pro-inflammatory cytokines in liver tissue of mice fed a high-fat diet (HFD) compared with those on a standard diet (SD). Indeed, we observed increased SA-β-gal staining (Supplementary Fig. 1g) and increased expression of senescence marker $p21^{waf1}$ (Supplementary Fig. 1h), as well as elevated levels of the cytokines monocyte chemoattractant protein 1 (MCP1), tumor necrosis factor α (TNFα) and interleukin 8 (IL8) (Supplementary Fig. 1i–k) in HFD-fed mice. Moreover, we detected the increased presence

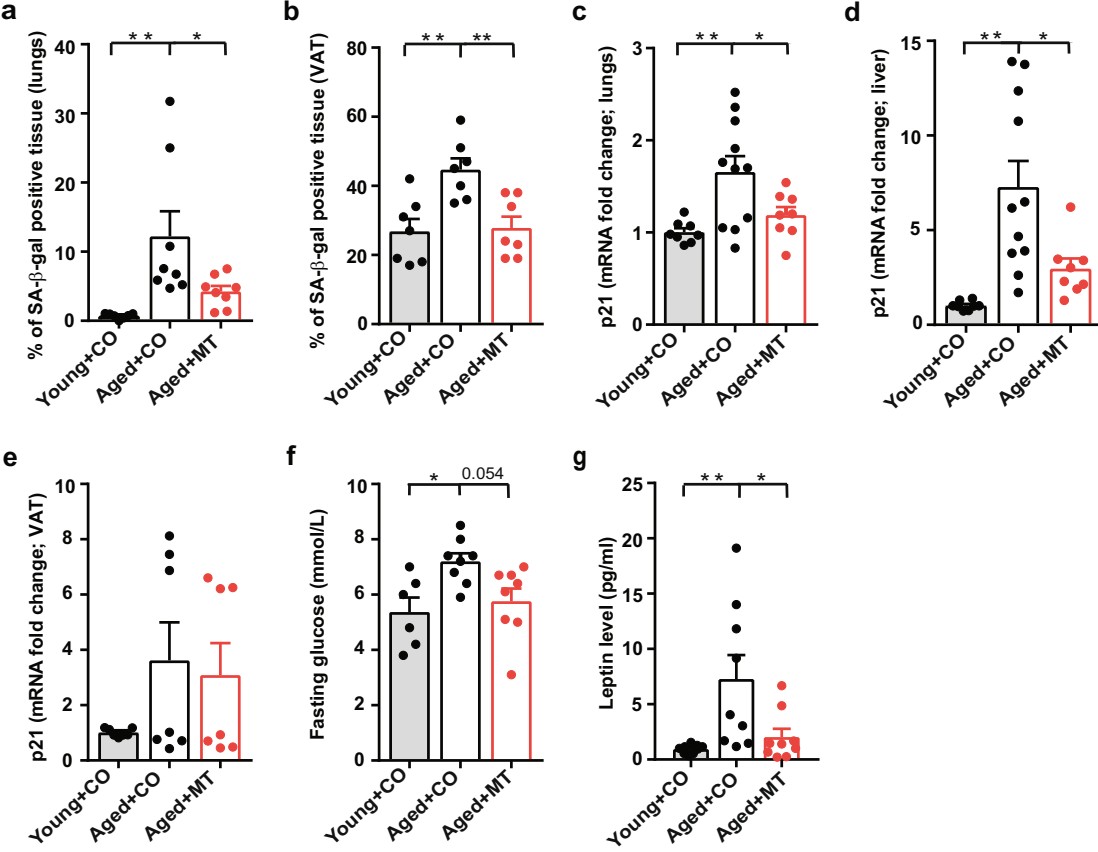

**Fig. 1 MitoTam improves metabolic parameters in naturally aged mice.** C57BL/6 mice 2 (young) and 18 months of age (aged) were treated i.p. once per week for a period of 4 weeks with MitoTam (2 mg/kg body weight; MT) dissolved in 4% EtOH in corn oil or with the vehicle (CO). **a, b** Lungs (**a**; young $n = 8$, aged $n = 10$, aged+MT $n = 8$; young vs. aged $p = 0.005$, aged vs. aged+MT $p = 0.049$) and VAT (**b**; $n = 7$; young vs. aged $p = 0.004$, aged vs. aged+MT $p = 0.006$) were excised and assessed for SA-β-gal activity, which was expressed as % of SA-β-gal-positive tissue. **c–e** Expression of $p21^{waf1}$ mRNA in lungs (**c**; young vs. aged $p = 0.005$, aged vs. aged+MT $p = 0.048$), liver (**d**; young vs. aged $p = 0.001$, aged vs. aged+MT $p = 0.035$; **c, d** young $n = 8$, aged $n = 11$, aged+MT $n = 8$) and VAT (**e**; $n = 7$) was estimated by qRT-PCR. **f** Fasting glucose level in young ($n = 6$), aged ($n = 8$) and aged mice treated with MitoTam ($n = 8$) was assessed after 16 h of fasting (young vs. aged $p = 0.021$, aged vs. aged+MT $p = 0.054$). **g** Level of leptin in plasma was estimated ($n = 9$; young vs. aged $p = 0.007$, aged vs. aged+MT $p = 0.026$). For all panels, **a–g** one-way ANOVA, Tukey´s comparison multiple test was used. Data are expressed as mean ± SEM; *$p < 0.033$; **$p < 0.002$; ***$p < 0.001$. Source data are provided as a Source Data file.

of senescence in adipose tissue of patients with obesity and T2DM (Supplementary Fig. 1l), which is consistent with the literature[15,23].

**MitoTam improves metabolic profile and attenuates senescence markers in mice-fed high-fat diet.** To see the effect of senescent cell elimination on obesity and prediabetes, we used the standard model of C57BL/6 male mice fed with HFD for 6 months[24,25] featuring significantly increased body weight and white adipose tissue mass compared to mice fed SD. This effect was reduced after treatment of HFD-fed mice with MitoTam for 4 weeks, which resulted in decreased body weight (Fig. 2a), weight of VAT and subcutaneous adipose tissue (SAT) (Fig. 2b; Supplementary Fig. 2b). Control HFD-fed mice had higher FI compared to SD-fed mice (17.69 ± 0.68 vs. 15.51 ± 0.67 kcal/day) (Fig. 2c). MitoTam treatment reduced FI in HFD-fed (12.86 ± 0.56 kcal/day; $P < 0.001$ vs. HFD + CO) but not in SD-fed mice (14.50 ± 0.95 kcal/day), thus, the effect of MitoTam on body and VAT weight reduction can be at least partly explained by lower FI (Fig. 2c).

A pair-feeding experiment was performed to determine the contribution of FI reduction to the overall effect of MitoTam on the physiology of HFD-fed mice. MitoTam-treated mice had lower FI (17.15 ± 0.58 vs. 20.31 ± 0.66 kcal/day) and the FI of

pair-fed (PF) mice was set to the same amount each day (17.20 ± 0.1 kcal/day) (Fig. 2d; Supplementary Fig. 2a). Despite the same FI, MitoTam-treated mice decreased their body weight and the weight of VAT higher than that for PF mice (Fig. 2e, f).

Decreased total body fat was accompanied by adipocyte size reduction in HFD-fed mice treated with MitoTam, while no effect of MitoTam was observed in SD-fed animals (Fig. 2g). HFD-fed mice showed elevated levels of senescence markers $p16^{Ink4a}$ and $p21^{waf1}$ mRNA, and of SA-β-gal activity in EAT (Fig. 2h, Supplementary Fig. 2c), which are closely related to chronic inflammation in the tissue. Most of these markers significantly decreased in response to MitoTam treatment in HFD- but not SD-fed animals or PF mice (Fig. 2i, Supplementary Fig. 2d), suggesting that MitoTam affects senescence by a different mechanism than only by reduced FI.

Brown adipocytes located mainly in SAT show the ability to transform from energy-storing to energy-dissipating tissue by activating expression of the mitochondrial uncoupling protein 1 (UCP1) (reviewed in ref. [26]). Increased expression of *UCP1* in SAT of HFD-fed animals treated with MitoTam indicates browning of white adipose tissue, however, no difference was observed between MitoTam and PF HFD-fed mice (Supplementary Fig. 2e, f). These results suggest that the effect of MitoTam on UCP1 was caused indirectly by reduced FI.

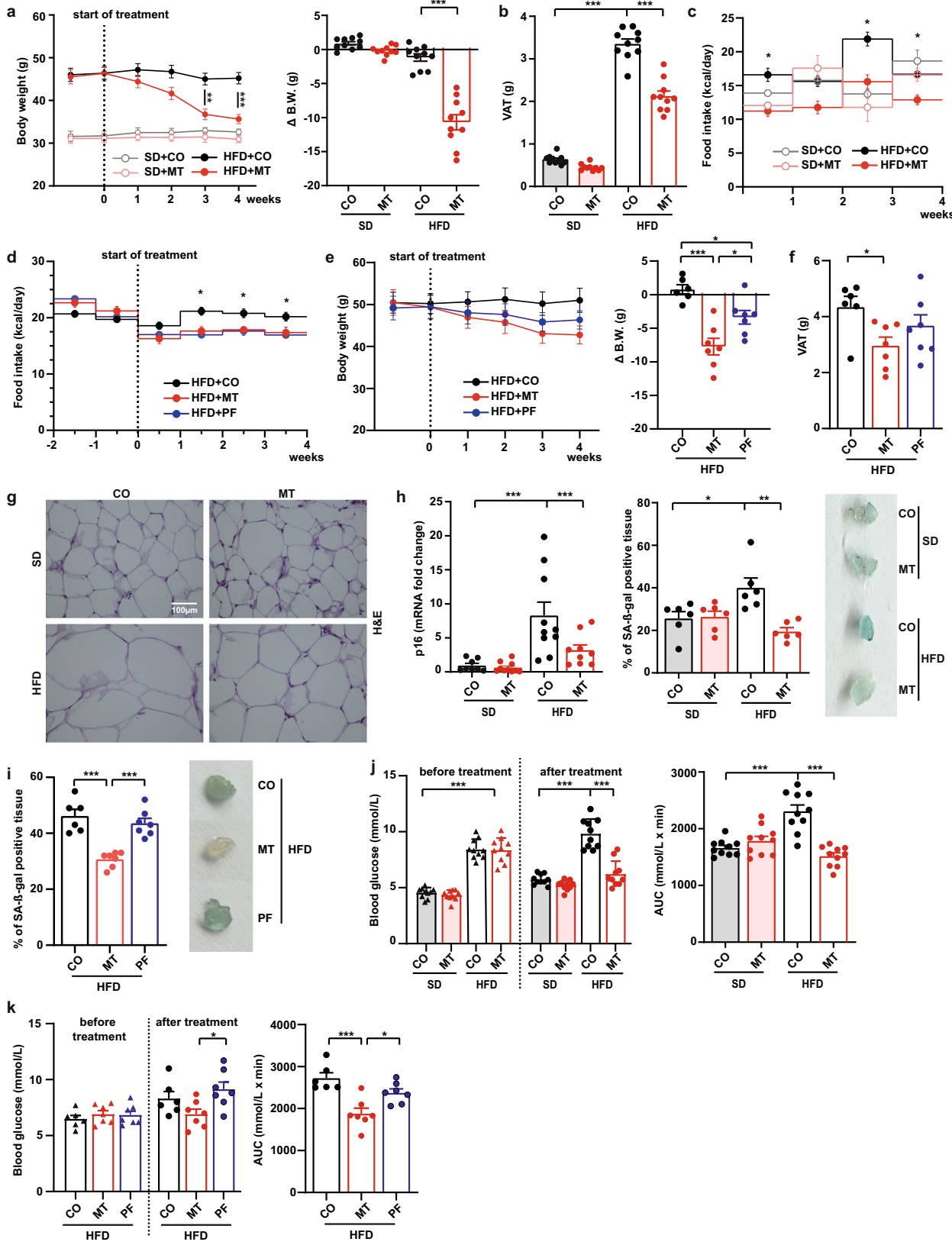

In addition to reducing body fat and body weight, MitoTam treatment reduced fasting glucose levels and improved glucose tolerance (oral glucose tolerance test (oGTT); expressed as area under curve, AUC) to the level of SD-fed mice (Fig. 2j), with significantly lower blood glucose values at all time points of the oGTT compared to control mice on HFD (Supplementary

Fig. 2g). Lowering of fasting blood glucose and oGTT improvement levels were not observed in PF mice (Fig. 2k, Supplementary Fig. 2h). The D-glucose levels in feces did not differ between MitoTam-treated and control mice (Supplementary Fig. 2i; mice fed ad libitum were used as a control) arguing against the role of impairment of intestinal glucose uptake in MitoTam's

**Fig. 2 MitoTam improves metabolic profile and attenuates senescence markers in mice fed high-fat diet.** C57BL/6 mice fed with SD (standard diet), HFD (high-fat diet) or HFD + PF (pair feed high-fat diet) were treated with MitoTam (MT; 2 mg/kg body weight dissolved in 4% ethanol in corn oil) or the vehicle (CO) given i.p. twice per week for a period of 4 weeks. **a, e** Total body weight was taken once per week throughout the experiment; (**a**; $n = 10$; HFD + CO vs. HFD + MT: 3rd week $p = 0.002$; 4th week $p < 0.001$), (**e**; HFD + CO $n = 6$, HFD + MT, HFD + PF $n = 7$ animals), Δ body weight was calculated as the difference between initial and final body weight; (**a**; HFD + CO vs. HFD + MT $p < 0.001$); (**e**; HFD + CO vs. HFD + MT $p < 0.001$, HFD + CO vs. HFD + PF $p = 0.032$, HFD + MT vs. HFD + PF $p = 0.019$). **b, f** Visceral adipose tissue (VAT) weight at the end of the experiment; (**b**; $n = 10$; SD + CO vs. HFD + CO $p < 0.001$, HFD + CO vs. HFD + MT $p < 0.001$), (**f**, HFD + CO $n = 6$; HFD + MT, HFD + PF $n = 7$; HFD + CO vs. HFD + MT $p = 0.040$). **c, d** FI was expressed in kcal/day per mouse; **c** ($n = 10$; 1st week HFD + CO vs. HFD + MT $p = 0.012$; 3rd week HFD + CO vs. HFD + MT $p = 0.0103$; 4th week HFD + CO vs. HFD + MT $p = 0.0146$), (**d**, HFD + CO $n = 6$, HFD + MT, HFD + PF $n = 7$; 2nd week HFD + CO vs. HFD + MT and HFD + PF $p = 0.0162$; 3rd week HFD + CO vs. HFD + MT and HFD + PF $p = 0.0296$; 4th week HFD + CO vs. HFD + PF $p = 0.0181$). **g** Histological samples of VAT were stained with H&E. **h** The level of mRNA of $p16^{Ink4a}$ in epididymal adipose tissue (EAT) was assessed by qRT-PCR (SD + CO $n = 8$; SD + MT, HFD + CO $n = 10$, HFD + MT $n = 9$; SD + CO vs. HFD + CO $p < 0.001$, HFD + CO vs. HFD + MT $p = 0.015$); % of SA-β-gal-positive tissue in EAT is shown as bar graph (left) and representative tissue is presented (right; blue color) ($n = 6$; SD + CO vs. HFD + CO $p = 0.024$, HFD + CO vs. HFD + MT $p = 0.001$). **i** % of SA-β-gal-positive tissue in EAT is shown as bar graph (left) and representative tissue is presented (right; blue color) (HFD + CO $n = 6$; HFD + MT, HFD + PF $n = 7$; HFD + CO and HFD + PF vs. HFD + MT $p < 0.001$). **j, k** Fasting blood glucose was determined at the beginning and at the end of the experiment, both after 16 h of fasting, and area under the oGTT curve (AUC) was estimated (**j**, $n = 10$; glucose before treatment SD vs. HFD $p < 0.001$, after treatment SD + CO vs. HFD + CO $p < 0.001$, HFD + CO vs. HFD + MT $p < 0.001$; AUC SD + CO vs. HFD + CO $p < 0.001$, HFD + CO vs. HFD + MT $p < 0.001$); (**k**, HFD + CO $n = 6$, HFD + MT, HFD + PF $n = 7$; glucose after treatment HFD + PF vs. HFD + MT $p = 0.032$; AUC HFD + CO vs. HFD + MT $p < 0.001$, HFD + PF vs. HFD + MT $p = 0.018$). For **a, e** (Δ B.W.), **b, f**, **h–k** One-way ANOVA, Tukey's comparison multiple test was used. For **a, e** (total B.W.), **c, d** Two-Way ANOVA, Tukey's comparison multiple test was used. All data are expressed as mean ± SEM; *$p < 0.033$; **$p < 0.002$; ***$p < 0.001$. Source data are provided as a Source Data file.

glucose-lowering effects. The glucose-lowering effect of MitoTam was accompanied by decreased fasting plasma triglycerides (TAG) and postprandial insulin, leptin, and gastric inhibitory polypeptide (GIP) levels (Fig. 3a) as well as decreased mRNA expression of *leptin* in EAT (Supplementary Fig. 2j) and insulin 1 (*INS1*) and insulin 2 (*INS2*) mRNA in the pancreas (Supplementary Fig. 2k). Similarly, the levels of TNFα and MCP1 in plasma tended to decrease in HFD-fed mice treated with MitoTam (Supplementary Fig. 2l).

Lipid accumulation in the liver promotes metabolic dysfunction and contributes to the development of T2DM and its further exacerbation. MitoTam reduced lipid accumulation in the liver of HFD-fed animals almost to the level found for SD-fed animals, as assessed by evaluation of TAG level, whereas in PF mice TAG level did not decrease (Fig. 3b). Histological evaluation of liver tissue indicated a pattern reminiscent of lowering of lipid accumulation in HFD-fed mice in response to MitoTam (Fig. 3e), which is documented by staining the liver tissue with Oil Red O (Fig. 3c, f), and further supported by collagen evaluation using Picrosirius red (Fig. 3d, g). Similarly, kidney tubular necrosis observed in HFD-fed mice was reduced after MitoTam treatment (Supplementary Fig. 2m). mRNA level of the senescence marker $p21^{waf1}$ (but not $p16^{Ink4a}$) was elevated in the liver of HFD-fed mice compared to the SD group and decreased significantly more in response to MitoTam treatment than in PF mice (Fig. 3h), which supports the previous results that reduced FI cannot fully explain elimination of senescent cells.

**MitoTam is superior to tamoxifen in the improvement of diabetic parameters.** Since MitoTam is a mitochondrially targeted analog of its parental compound tamoxifen, we next explored whether mitochondrial targeting is essential for its activity presented in Figs. 1–3, comparing the effects of MitoTam and tamoxifen.

Mice fed HFD for 15 months showed increased body weight that was reduced by 4-week treatment with MitoTam as well as tamoxifen (Fig. 4a). VAT (including its components EAT and PRAT), as well as SAT weight, decreased more in response to MitoTam than to tamoxifen treatment (Fig. 4b, Supplementary Fig. 3a). FI was slightly lower in MitoTam-treated mice with a significant difference during the 2nd and 3rd week of treatment (Fig. 4c). Compared to the baseline, fasting blood glucose was

raised by 22% in control mice and by 12% in mice treated with tamoxifen, whereas it decreased by 16.5% in MitoTam-treated animals (Fig. 4d). Although the AUC calculated from oGTT values (Supplementary Fig. 3b) was reduced in both treated groups, MitoTam was more effective (Fig. 4d). Tamoxifen had no significant effect on insulin and GIP levels, while MitoTam significantly decreased both parameters (Fig. 4e). Fasting plasma TAG showed a slight reduction after MitoTam administration (Supplementary Fig. 3c).

Despite decreased VAT weight in both treated groups, circulating leptin levels and its mRNA in EAT decreased in response to MitoTam but not tamoxifen (Fig. 4f), suggesting a more pronounced metabolic effect of MitoTam on adipose tissue. SA-β-gal staining of EAT showed a profound reduction of senescent cells in MitoTam-treated mice (Fig. 4g). These data were accompanied by a slight decrease of $p16^{Ink4a}$ level after MitoTam treatment (Fig. 4g). HFD-fed mice showed markedly increased lipid accumulation in the liver (Fig. 4h–j) and kidney tubular necrosis (Fig. 4k), which was reduced by MitoTam considerably more than by tamoxifen. Liver TAG was reduced in both treated groups compared to control mice (Supplementary Fig. 3d), while $p21^{waf1}$ mRNA level tended to be suppressed by MitoTam but not by tamoxifen (Supplementary Fig. 3e).

**Persistence of MitoTam effect in HFD-fed mice.** To see whether the effect of MitoTam on metabolic parameters persists, we evaluated relevant markers after cessation of the treatment. C57BL/6 mice fed either HFD or SD were treated with MitoTam or the vehicle, and metabolic parameters were monitored immediately after the last dose of the agent and 1 month later, except for body weight monitored on weekly basis for 7 weeks. Data collected immediately at the end of the treatment confirmed the above results with improved glucose uptake (IPG; expressed as area under the curve, AUC), loss of body weight, and decreased fasting glucose in MitoTam-treated HFD-fed mice (Fig. 5a–c, Supplementary Fig. 4a). Compared to these results, cessation of treatment led to the shift of glucose concentrations and glucose tolerance back to levels seen in control HFD-fed mice (Fig. 5a, c, Supplementary Fig. 4a). Interestingly, the body weight of HFD-fed animals remained reduced for 7 weeks after the treatment cessation (Fig. 5b), which was associated with the decreased total body fat (Fig. 5d, e, Supplementary Fig. 4b), while lean

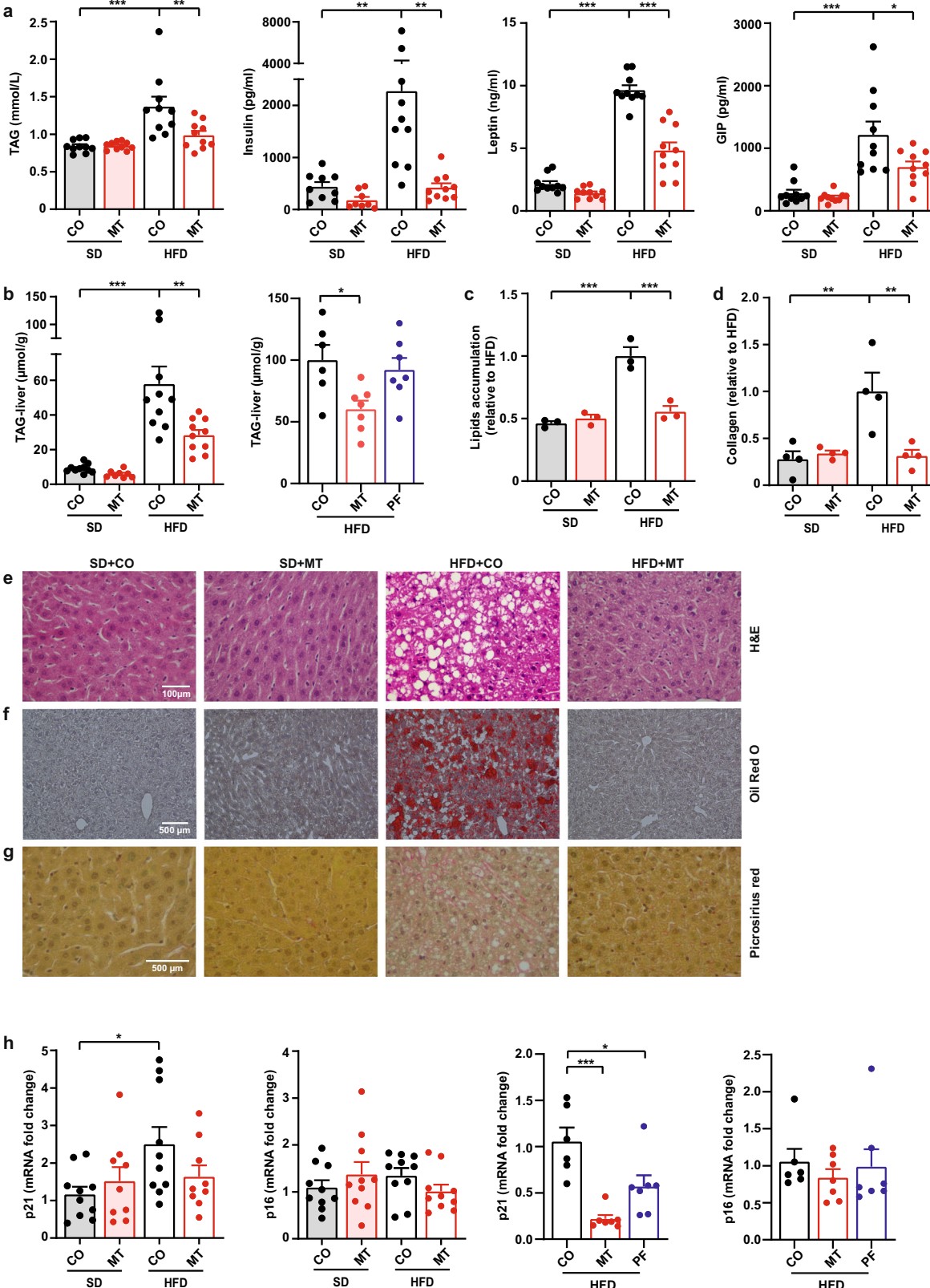

body mass (Fig. 5f) and organ weight were unchanged (Supplementary Fig. 4f).

Similarly, serum insulin concentrations and its mRNA level in the pancreas remained lowered even 1 month after the end of treatment (Fig. 5g, Supplementary Fig. 4c). As shown previously, plasma leptin levels in HFD-fed mice declined in response to MitoTam. However, the same reduction was not observed 1 month after the end of treatment, even though leptin expression in VAT remained reduced (Fig. 5h, Supplementary Fig. 4d). Importantly, considerably decreased lipid accumulation in the liver of HFD-fed mice treated with MitoTam was detected even 1 month after the treatment cessation (Fig. 5i), which is

**Fig. 3 MitoTam improves metabolic profile and attenuates senescence markers in mice fed high-fat diet.** C57BL/6 mice fed with SD (standard diet), HFD (high-fat diet) or HFD + PF (pair feed high-fat diet) were treated with MitoTam (2 mg/kg body weight dissolved in 4% ethanol in corn oil; MT) or the vehicle (CO) given i.p. twice per week for a period of 4 weeks. **a** Fasting plasma triglycerides (TAG) and postprandial serum insulin, leptin, and gastric inhibitory polypeptide (GIP) concentration were assessed; TAG ($n = 10$; TAG: SD + CO vs. HFD + CO $p < 0.001$, HFD + CO vs. HFD + MT $p = 0.004$), Leptin ($n = 10$; SD + CO and HFD + MT vs. HFD + CO $p < 0.001$), GIP SD + CO vs. HFD + CO $p < 0.001$, HFD + CO vs. HFD + MT $p = 0.02$), Insulin (SD + CO $n = 9$; SD + MT $n = 8$; HFD + CO, HFD + MT $n = 10$; SD + CO vs. HFD + CO $p = 0.003$, HFD + CO vs. HFD + MT $p = 0.002$). **b** Levels of TAG in the liver were evaluated (SD + CO, HFD + CO, HFD + MT $n = 10$; SD + MT $n = 8$; SD + CO vs. HFD + CO $p < 0.001$; HFD + CO vs. HFD + MT $p = 0.003$; PF experiment: HFD + CO $n = 6$, HFD + MT, HFD + PF $n = 7$; HFD + CO vs. HFD + MT $p = 0.026$). **c, f** Oil Red O staining was used for lipid accumulation in the liver (averages from four independent measurements of three samples; SD + CO and HFD + MT vs. HFD + CO $p < 0.001$). **d, g** Picrosirius red staining of total collagen in the liver was performed (averages from three independent measurements of four samples; SD + CO vs. HFD + CO $p = 0.004$; HFD + CO vs. HFD + MT $p = 0.005$). **e** Histology of liver sections is shown by H&E staining. **h** $p21^{waf1}$ and $p16^{Ink4a}$ mRNA levels in the liver were assessed by qRT-PCR ($p21^{waf1}$: SD + CO, HFD + CO $n = 10$, SD + MT, HFD + MT $n = 9$; SD + CO vs. HFD + CO $p = 0.0409$; $p16^{Ink4a}$: $n = 10$; PF experiment: $p21^{waf1}$ and $p16^{Ink4a}$: HFD + CO $n = 6$, HFD + MT, HFD + PF $n = 7$; $p21^{waf1}$: HFD + CO vs. HFD + MT $p < 0.001$, HFD + CO vs. HFD + PF $p = 0.018$). One-Way ANOVA; Tukey's comparison multiple test. All data are expressed as mean ± SEM *$p < 0.033$; **$p < 0.002$; ***$p < 0.001$. Source data are provided as a Source Data file.

supported by decreased TAG accumulation in the liver (Supplementary Fig. 4e).

The effect of MitoTam on energy metabolism was studied by indirect calorimetry. HFD-fed MitoTam-treated mice manifested lower total energy expenditure (TEE) at the end and 1 month after treatment compared to HFD-fed controls (Supplementary Fig. 5a, b). The respiratory exchange ratio (RER) of HFD-fed mice was lower, reflecting the diet composition, but unaffected by MitoTam (Supplementary Fig. 5c, d). No significant effect of MitoTam on FI was detected, probably due to high data variability caused by short measurement time (24 h) inside the calorimetry chamber (Supplementary Fig. 5e, f). MitoTam did not affect physical activity (PA) at the end and 1 month after the treatment (Supplementary Fig. 5g, h). To assess whether the TEE reduction corresponds to lower body weight of MitoTam-treated mice, TEE to body weight regressions were constructed, using all mouse data to reflect predicted TEE changes with changed body weight (Supplementary Fig. 5i). HFD MitoTam-treated mouse data were equally distributed around this line at the end of the treatment and were all below this level 1 month after the treatment. Thus, the TEE reduction at the end of the treatment can be explained by lower body weight, while some other factors contributed to the TEE reduction 1 month after the treatment cessation (Supplementary Fig. 5i). The cardiac function was not negatively affected by MitoTam (Supplementary Fig. 5j, k). Overall, the indirect calorimetry data show that the mechanism of MitoTam-induced body weight reduction was not due to higher energy expenditure.

**MitoTam suppresses differentiation of pre-adipocytes and rejuvenates adipose tissue.** Impairment of mitochondrial function in white adipose tissue is strongly connected to the development of chronic inflammation, senescence, and T2DM. Thus, MitoTam treatment led to improvement of T2DM-related parameters with reduction of adipose tissue mass and associated decrease in deposition of lipids in various tissues of HFD-fed mice. Therefore, clarification of the mechanism by which Mito-Tam positively affects adipose tissue function is of interest. For this purpose, we used a plausible model presented by 3T3-L1 pre-adipocyte cells, which can be differentiated into mature adipocytes using a "cocktail" of adipogenesis inducers.

We first confirmed the senolytic effect of MitoTam on mature senescent adipocytes (Fig. 6a) as evidenced by increased death of treated cells (Fig. 6b, c), confirmed by the decrease of lipid accumulation in cell culture (Fig. 6d, e), production of TAG into the medium (Fig. 6f) and decreased level of SA-β-gal (Fig. 6g, h).

As the effect of MitoTam on HFD-fed mice was pronounced in the adipose tissue of mice, we assumed that elimination of

senescent cells may not be the only possible explanation for the mechanism of action of this drug on obesity and T2DM. Adipogenesis has been shown to affect the composition and regeneration of adipose tissue during obesity[27]. Since adipocyte differentiation occurs in two steps[28,29], MitoTam and tamoxifen were added to pre-adipocytes at two different time points (Fig. 7a), and evaluations were performed after completion of adipogenesis (day 9). Untreated and tamoxifen-treated adipocytes accumulated TAG in the form of lipid bodies (Fig. 7b, c), pointing to completed adipocyte differentiation. Lipid accumulation was almost fully suppressed by MitoTam when the cells were treated in the early stage of adipogenesis (day 0–2) (Fig. 7b, c). Moreover, the effect of MitoTam on TAG accumulation correlated with its concentration and time of administration (Supplementary Fig. 6a–d). In subsequent experiments, MitoTam was used at 0.32 μM, which is comparable to our previous in vivo experiments and which was non-toxic over the 9 days of adipogenesis.

To further verify the effect of MitoTam on adipogenesis, we quantified the expression of Ppar-γ and Cebp-α transcription factors that regulate differentiation of pre-adipocytes as well as the relevant adipokines adiponectin (AdipoQ) and leptin. As expected, we detected increased expression of all four genes in fully differentiated, untreated adipocytes. In contrast, little or no increase of transcripts of these genes was detected following MitoTam administration in both stages of adipogenesis (Fig. 7d–g), whereas tamoxifen was inefficient. Furthermore, we observed decreased expression of Ppar-γ and Cebp-α in EAT of HFD-fed mice treated with MitoTam (Supplementary Fig. 6e, f). Tamoxifen treatment did not affect the expression of any of these genes. Proteomic analysis of 3T3-L1 cells revealed a shift of the proteome of MitoTam-treated cells to its pattern in pre-adipocytes, while tamoxifen-treated cells did not show any changes when compared to mature adipocytes (Supplementary Fig. 6g). These results document that MitoTam, but not tamoxifen, inhibits adipogenesis, especially when applied during the early stages of the process.

Within the first 24 h after initiation of adipogenesis, cells undergo several rounds of Ppar-γ- and Cebp-α-dependent mitosis, referred to as clonal mitotic expansion[30–32]. Since MitoTam decreased expression of both genes, we tested its effect on cellular proliferation. When evaluated after the first 2 days of differentiation, untreated and tamoxifen-treated pre-adipocytes showed an approximately threefold increase in cell number, while the number of cells remained unchanged upon MitoTam treatment (Fig. 7h).

Mitochondrial bioenergetics and organization within 3T3-L1 cells have been reported to be altered during adipogenesis. It changes from the continuous reticulum to fragmented

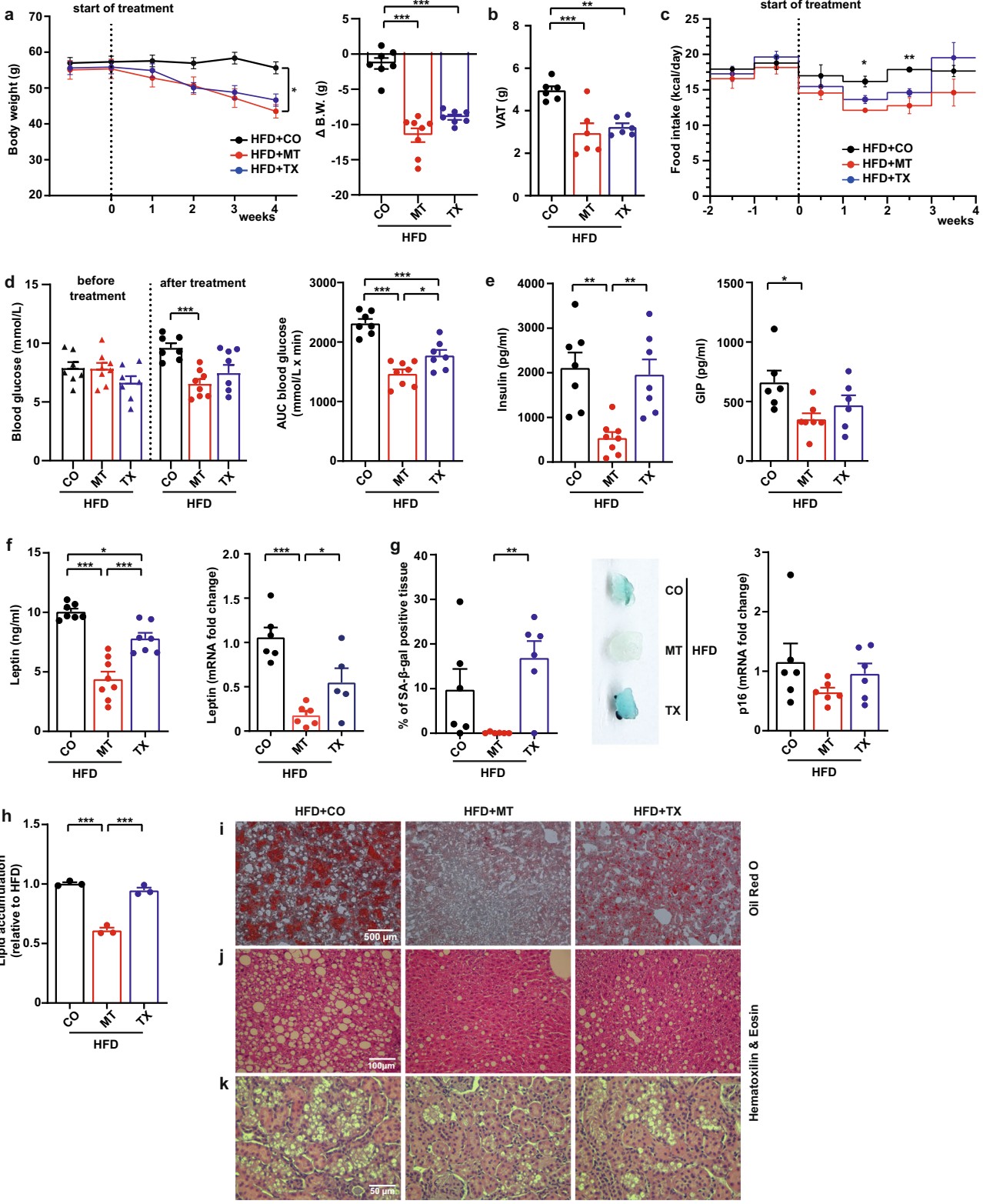

mitochondria[33] with increased mitochondrial activity indicated by higher oxygen consumption[34]. In addition to previous experiments, MitoTam, but not tamoxifen treatment prevented changes in the mitochondrial network during differentiation, especially when added at the beginning of the process (Fig. 7i). Oxygen consumption of adipocytes further supports the above data, as it was highly increased in non-treated cells but not in pre-adipocytes treated with MitoTam within the first 2 days of

differentiation (Fig. 7j). Moreover, fluorescence lifetime imaging microscopy (FLIM) revealed a negative shift in relative levels of free/bound NADH ratio during 3T3-L1 adipogenesis, indicating more NADH bound to respiratory complex I (Fig. 7k). Interestingly, accumulation of free NADH, observed during decreased respiration, did not apply for cells treated with MitoTam during the first two days of differentiation. Increased binding of NADH reflects uncoupled respiration of these cells detected following the

**Fig. 4 MitoTam is superior to tamoxifen in improvement of diabetic paramtepers.** HFD-fed C57BL/6 mice were divided into three groups: HFD + CO (corn oil); HFD + MT (MitoTam); HFD + TX (tamoxifen); the animals were treated with MitoTam or tamoxifen (2 mg/kg body weight) dissolved in 4% ethanol in corn oil or the vehicle given i.p. twice per week for a period of 4 weeks. **a** Total body weight was taken once per week (HFD + CO, HFD + TX $n = 7$; HFD + MT $n = 8$; 4th week: HFD + CO vs. HFD + MT $p = 0.019$); Δ body weight was calculated as difference between the initial and final body weight (HFD + CO, HFD + TX $n = 7$; HFD + MT $n = 8$; HFD + CO vs. HFD + MT $p < 0.001$, HFD + CO vs. HFD + TX $p < 0.001$). **b** Visceral adipose tissue (VAT) weight was taken at the end of the experiment ($n = 6$; HFD + CO vs. HFD + MT $p < 0.001$, HFD + CO vs. HFD + TX $p = 0.003$). **c** FI expressed in kcal/day per mouse (HFD + CO, HFD + TX $n = 7$; HFD + MT $n = 8$; 2nd week: HFD + CO vs. HFD + MT $p = 0.037$; 3rd week: HFD + CO vs. HFD + TX $p = 0.0033$). **d** Fasting blood glucose at the beginning and end of the experiment was evaluated after 16 h of fasting (HFD + CO, HFD + TX $n = 7$; HFD + MT $n = 8$; after treatment HFD + CO vs. HFD + MT $p < 0.001$) and area under the oGTT curve (AUC) were estimated (HFD + CO, HFD + TX $n = 7$; HFD + MT $n = 8$; HFD + CO vs. HFD + MT $p < 0.001$, HFD + CO vs. HFD + TX $p < 0.001$, HFD + MT vs. HFD + TX $p = 0.029$). **e** Postprandial serum insulin and gastric inhibitory polypeptide (GIP) concentrations were assessed (insulin: HFD + CO, HFD + TX $n = 7$; HFD + MT $n = 8$; HFD + CO vs. HFD + MT $p = 0.002$, HFD + TX vs. HFD + MT $p = 0.005$; GIP: HFD + CO $n = 6$, HFD + MT $n = 7$, HFD + TX $n = 6$; HFD + CO vs. HFD + MT $p = 0.029$). **f** Postprandial serum leptin protein and mRNA in epididymal adipose tissue (EAT) were evaluated (HFD + CO, HFD + TX $n = 7$; HFD + MT $n = 8$; serum leptin: HFD + CO vs. HFD + MT $p < 0.001$, HFD + CO vs. HFD + TX $p = 0.014$, HFD + TX vs. HFD + MT $p < 0.001$; *leptin* mRNA: HFD + CO vs. HFD + MT $p = 0.0043$, HFD + MT vs. HFD + TX $p = 0.0321$). **g** % of SA-β-gal-positive tissue in EAT is shown as bar graph (left), and representative tissue is presented (right; blue color) ($n = 6$; HFD + TX vs. HFD + MT $p = 0.010$); $p16^{Ink4a}$ mRNA levels in EAT were assessed by qRT-PCR ($n = 6$). **h, i** Oil Red O staining was used for lipid accumulation (averages from 4 independent measurements of three samples; HFD + CO vs. HFD + MT $p < 0.001$, HFD + TX vs. HFD + MT $p < 0.001$). **j, k** Histology of liver sections is shown by H&E staining. For **a** (Δ B.W.), **b**, **d**–**g** One-Way ANOVA, Tukey´s comparison multiple test was used. For **a** (total B.W.), **c** Two-Way ANOVA, Tukey´s comparison multiple test was used. All data are expressed as mean ± SEM; $*p < 0.033$; $**p < 0.002$; $***p < 0.001$. Source data are provided as a Source Data file.

addition of oligomycin A (Fig. 7j), which is in line with the slight drop of mitochondrial potential (Supplementary Fig. 6h). However, changes in respiration had no effect on the viability of MitoTam-treated cells (Supplementary Fig. 6i).

Escalation of oxygen consumption in fully differentiated adipocytes triggers adenine nucleotide transporter-2 (ANT2)-dependent stabilization of hypoxia-inducible factor 1α (HIF1α) and development of hypoxia[35]. Nevertheless, the expected rise of *ANT2* and *HIF1α* expression in mature adipocytes was reduced after MitoTam, but not tamoxifen treatment in vitro (Supplementary Fig. 6j) and in vivo (Supplementary Fig. 6k, l). Hypoxia is also closely related to the development of senescence. Hence, MitoTam application in both stages of adipogenesis led to a reduction of senescence markers, observed in mature adipocytes (Fig. 7l, m, Supplementary Fig. 6m).

## Discussion

T2DM represents a major health threat due to increasing prevalence related to the worldwide spread of obesity and debilitating long-term chronic complications, in a particularly increased risk of cardiovascular events, diabetic kidney disease, liver steatosis, diabetic neuropathy, and retinopathy[36]. The presence of diabetic complications is the major reason for the massive financial burden of the treatment of T2DM, and it is estimated that therapy of diabetic complications consumes up to two-thirds of the overall diabetes treatment costs. Despite the availability of novel glucose-lowering drugs with positive effects on the development of cardiovascular complications of diabetes and diabetic kidney disease[37,38], the number of patients with T2DM and accompanied chronic complications keeps increasing at a high rate.

Current pharmacological approaches to the treatment of T2DM are focused on enhancing insulin secretion, improving insulin sensitivity, or decreasing blood glucose levels by attenuating its renal re-absorption[39,40]. These treatment modalities address the pathophysiological defects present in T2DM rather than preventing the processes contributing to its evolution and development. Accumulation of senescent cells has been linked to obesity and the development of T2DM and its complications via impeding the regeneration and maintenance of renewable tissues and contributing to tissue aging[9]. Premature senescence may, therefore, represent an important and as thus far undervalued mechanism preceding T2DM development and contributing to insulin resistance, β-cell dysfunction, and apoptosis[13].

Here we show that treatment with MitoTam, an anticancer agent with senolytic activity[18], markedly improved glucose control in the HFD mouse model of obesity and prediabetes mimicking patients with diabetes, whose excessive calorie intake leads primarily to obesity and insulin resistance with the gradual development of hyperglycemia representing a typical time-course for the development of human T2DM[24,25]. This was accompanied by reduction of senescent cells and alleviation of diabetes-related comorbidities epitomized by liver steatosis and dyslipidemia. The improvement of glucose control after MitoTam administration was associated with decreased insulin levels, suggesting that improved insulin sensitivity rather than changes in insulin secretion can explain MitoTam effects. Moreover, specific mitochondrial targeting of MitoTam allows the agent to be used at considerably lower doses compared to metformin or to a combination of desatinib with quercetin tested in clinical trials as senolytic agents with a confirmed role in the improvement of T2DM[14,41,42].

Diabetes-related complications comprise the most significant burden of T2DM as its treatment is costly and often unsuccessful. Increased fat accumulation in the liver, referred to as NAFLD, presents typical comorbidity in patients with T2DM, which can progress into non-alcoholic steatohepatitis and further to liver cirrhosis, a common cause of liver failure requiring its transplantation. Similar to T2DM, NAFLD incidence increases with age. There is evidence that senescent hepatocytes are a major driver of liver steatosis, possibly via the inability of mitochondria to efficiently metabolize fatty acids with subsequent excessive lipid accumulation in the liver. In support of this, the elimination of senescent cells attenuates the progression of NAFLD[16]. Our data show that elimination of senescent cells by MitoTam treatment in mice with diet-induced obesity and prediabetes reduced excessive lipid accumulation in the liver almost to levels comparable to SD-fed animals along with decreased collagen, a marker of liver fibrosis. As ectopic lipid accumulation in the liver closely correlates with insulin resistance and its decrease improves insulin sensitivity, our data suggest that increased liver insulin sensitivity after MitoTam administration could be in part responsible for its glucose-lowering effects.

Obesity is connected to metabolic complications such as T2DM and dyslipidemia as well as increased risk of cancer and cardiovascular morbidity and mortality. MitoTam treatment significantly reduced body weight and decreased adipose tissue mass in mice with diet-induced obesity and prediabetes, which

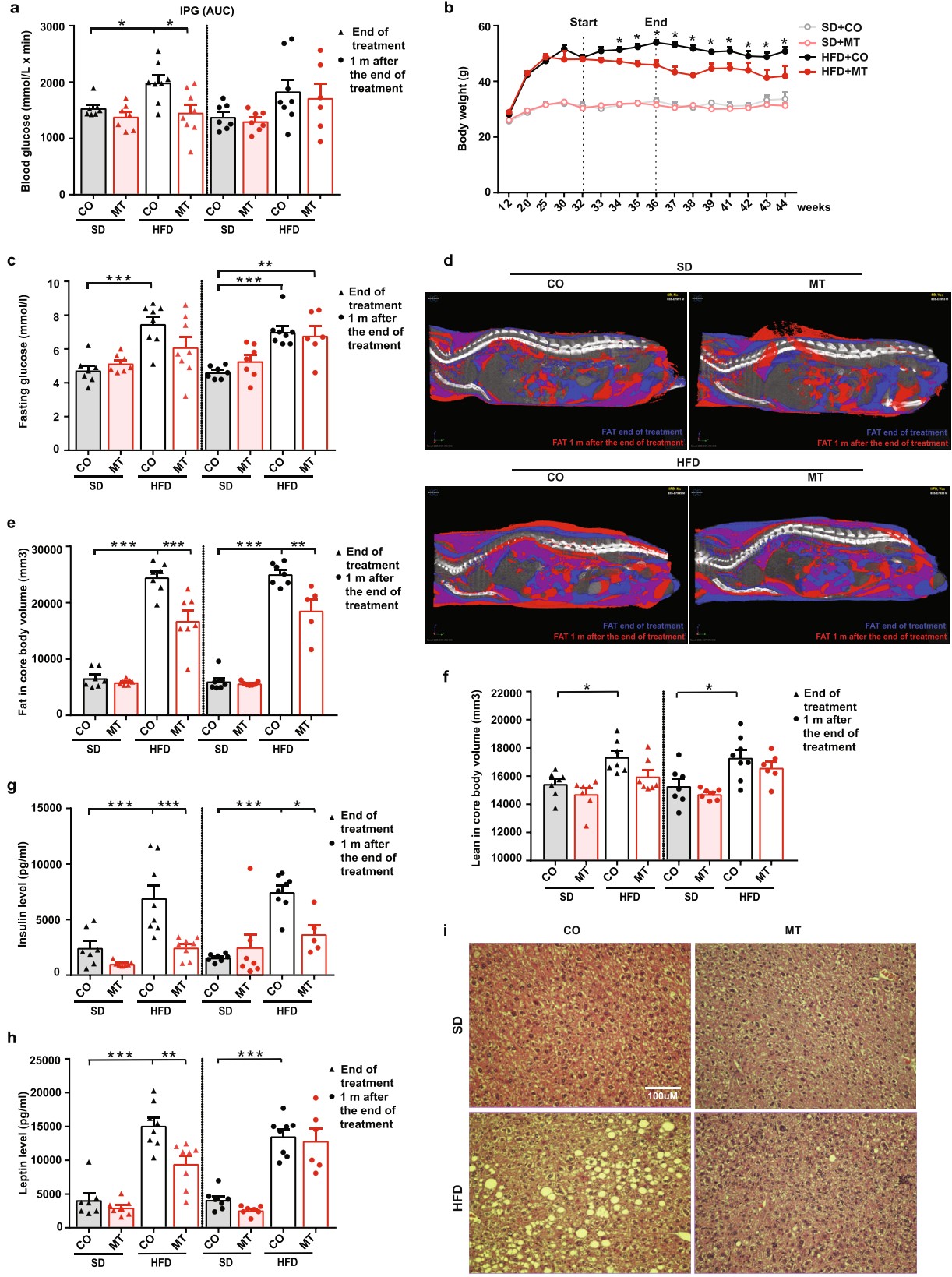

could have contributed to both improved glucose control and attenuated fatty liver disease. Although MitoTam treatment did not affect energy expenditure assessed by indirect calorimetry, decreased FI after MitoTam treatment was observed in animals fed HFD. Together with the elimination of hypertrophic adipocytes and decreased level of TAG as well as lipid deposition and

decreased intestinal absorption related to deteriorated metabolism, this is one of the important players in overall changes in body weight, which will be evaluated in further experiments.

The role of tamoxifen in the improvement of glycemia and fat reduction was described in the literature[43]. Consistent with this, the effect of tamoxifen is strongly connected with activation of the

**Fig. 5 Prolonged effect of MitoTam on mice with impaired metabolism.** C57BL/6 mice fed either with HFD or SD were treated i.p. twice per week for a period of 4 weeks with MitoTam (2 mg/kg body weight; MT) dissolved in 4% EtOH in corn oil or with the vehicle (CO). **a** Intraperitoneal glucose tolerance test (IPG) was performed at the end and 1 month after the end of the treatment, and area under the curve (AUC) was calculated (end of treatment SD $n = 7$, HFD $n = 8$; 1 m after treatment SD + CO, SD + MT $n = 7$, HFD + CO $n = 8$, HFD + MT = 6; SD + CO vs. HFD + CO $p = 0.0423$, HFD + CO vs. HFD + MT $p = 0.0101$). Body weight was measured weekly, starting at 12 weeks of age, ending 7 weeks after treatment termination (**b**; $n = 8$). Fasting blood glucose, assessed after 16 h of fasting, was determined at the end and 1 month after the end of treatment (**c**; end of treatment SD + CO, SD + MT $n = 7$, HFD + CO, HFD + MT $n = 8$; 1 m after treatment SD + CO, SD + MT $n = 7$, HFD + CO $n = 8$, HFD + MT = 6; SD + CO vs. HFD + CO both $p = 0.001$, SD + CO vs. HFD + MT 1 m after the end of treatment $p = 0.002$). Area of total fat in core body (mm$^3$) was determined and 3D visualization was performed at the end and 1 month after the treatment cessation (**d** representative images; **e** end of treatment $n = 7$; 1 m after treatment SD + CO, SD + MT = 7, HFD + CO $n = 7$, HFD + MT = 5; SD + CO vs. HFD + CO both, HFD + CO vs. HFD + MT end of treatment $p = <0.001$, HFD + CO vs. HFD + MT 1 m after the end of treatment $p = 0.003$). Each image represents overlap of two scans (before treatment and 1 month after the end of treatment, scans were superimposed based on the position of mouse backbone and ribs), purple color represents overlap of adipose tissue present in the mouse during both scans. Blue color represents adipose tissue detected only during scanning before treatment, red color reveals adipose tissue detected only 1 month after the treatment. Area of total lean in core body (mm$^3$) was determined at the end and 1 month after the treatment cessation (**f** end of treatment SD $n = 7$, HFD $n = 8$, SD + CO vs. HFD + CO $p = 0.028$; 1 m after treatment SD + CO, SD + MT = 7, HFD + CO $n = 8$, HFD + MT = 6; SD + CO vs. HFD + CO $p = 0.027$). Insulin (**g** end of treatment SD $n = 7$, HFD $n = 8$; 1 m after treatment SD + CO, SD + MT $n = 7$, HFD + CO $n = 8$, HFD + MT = 5; SD + CO vs. HFD + CO both, HFD + CO vs. HFD + MT end of treatment $p = <0.001$, HFD + CO vs. HFD + MT 1 m after the end of treatment $p = 0.019$) and leptin (**h** end of treatment SD $n = 7$, HFD $n = 8$; 1 m after treatment SD + CO, SD + MT $n = 7$, HFD + CO $n = 8$, HFD + MT = 6; SD + CO vs. HFD + CO both $p = <0.001$, HFD + CO vs. HFD + MT end of treatment $p = 0.003$) blood levels were assessed immediately and 1 month after treatment termination. **i** Lipid accumulation in the liver was visualized in histological samples stained with haematoxylin and eosin 1 month after the end of the treatment. One-way ANOVA, Tukey's comparison multiple test was used. Data are expressed as mean ± SEM; *$p < 0.033$; **$p < 0.002$; ***$p < 0.001$. Source data are provided as a Source Data file.

estrogen receptor (ER). In our previous experiments[19], we observed the same effect of MitoTam on the elimination of cancer cells independently of their ER status. Similarly, we did not observe any ER activation after MitoTam treatment in the liver of HFD-fed mice compared to tamoxifen indicating that our modification of tamoxifen with TPP changed its mode of action to be ER-independent. Importantly, unlike tamoxifen, which only decreased hyperglycemia and reduced adipose tissue mass, MitoTam showed a more complex effect on the improvement of diabetic parameters including the elimination of senescent cells, the alleviation of T2DM comorbidities, or suppression of systemic inflammation. Furthermore, we observed a prolonged effect of MitoTam on weight reduction in comparison with tamoxifen, where only transient effect followed by over-compensation resulting in increased fat mass and development of T2DM was described[44–46]. This makes MitoTam more favorable than other currently used therapies that require daily administration with doses higher by one or more orders of magnitude[41,47–49]. Moreover, most of these therapies only decrease FI, while MitoTam targets a key mechanistic property, indicating its clinical potential in combination with altered lifestyle.

It is known that mitochondrial stress leads to activation of the integrated stress response via activation of transcription factor 4, resulting in increased expression of mitokines such as fibroblast growth factor 21 (FGF21) and growth differentiation factor 15 (GDF15), which both have been shown to improve diabetes-related features (reviewed in ref. [50]). Analyzing the expression of *FGF21* and *GDF15* genes in tissues as well as the level of FGF21 in plasma, we have confirmed the role of FGF21 in diabetes induction/maintenance, but this mitokine is not essential for the effect of MitoTam, since there was no change in its expression after treatment despite improved diabetic parameters observed in treated animals (Supplementary Fig. 7).

Most of the factors affecting progression of T2DM are tightly connected with adipose tissue (e.g., increased expression of pro-inflammatory factors or hormones, which can induce cellular damage or senescence in distal organs such as the liver or pancreas). Since the highest level of senescent cells was described particularly in adipose tissue[14], changes in this tissue that include not only a loss of adipose tissue mass but mainly functional changes, therefore represent an important mechanism in the regulation of diabetic parameters. During adipogenesis, pre-adipocytes undergo sequential transcriptional regulation by adipogenic regulators such as the peroxisome proliferator-activated receptor-γ (PPAR-γ), CCAAT-enhancer-binding proteins (C/EBPs), and PPARγ coactivator 1-α (PGC1-α), which leads to promotion of mitochondrial biogenesis[51] as well as production of ATP required for increased metabolic activity during adipogenesis[52]. Decreased expression of adipogenic regulators together with decreased mitochondrial expansion and triglyceride accumulation in adipocytes indicate that MitoTam attenuates adipogenesis. Our results are supported by previous work showing that knocking out mitochondrial transcription factor TFAM, a key regulator of transcription of mitochondria-encoded genes and of mitochondrial replication and expansion, resulted in the protection of mice from age- and diet-induced obesity, glucose intolerance, and liver steatosis via increased energy expenditure, specifically in adipose tissue[53]. Since insulin and leptin, acting as C/EBPβ and PPARγ regulators, play an important role in senescence induction[54,55], we propose that their decreased production after MitoTam treatment represents a negative loop in the regulation of pre-adipocyte differentiation.

In summary, we propose a combined effect of MitoTam on different tissues leading together to the improvement of obesity and T2DM-related parameters. Elimination of lipid-accumulating senescent cells in MitoTam-treated mice suggests that the underlying mechanism(s) could be linked to attenuation of pre-mature senescence that has been implicated in the pathogenesis of T2DM[9]. This could open a new avenue of prevention and/or treatment of T2DM and its comorbidities. At the moment, glucose-lowering medications generally target insulin or glucagon secretion (sulphonylureas, incretin-based therapies), insulin deficiency (replacement with exogenous insulin), insulin resistance (metformin, pioglitazone), or increased glucose elimination via urine (sodium-dependent glucose transporter-2 inhibitors)[3]. None of the available treatment options is based on the elimination of the excessive presence of senescent cells in adipose tissue or other vital organs. It is therefore plausible to expect that MitoTam, possibly in combination with other glucose-lowering therapies, will bring complementary and complex effects on both glucose control and chronic diabetes complications, as we show in our work for liver steatosis and kidney damage.

To conclude, MitoTam represents a type of as yet untested anti-diabetic agent contributing to curbing one of the most widespread pandemics in industrialized countries.

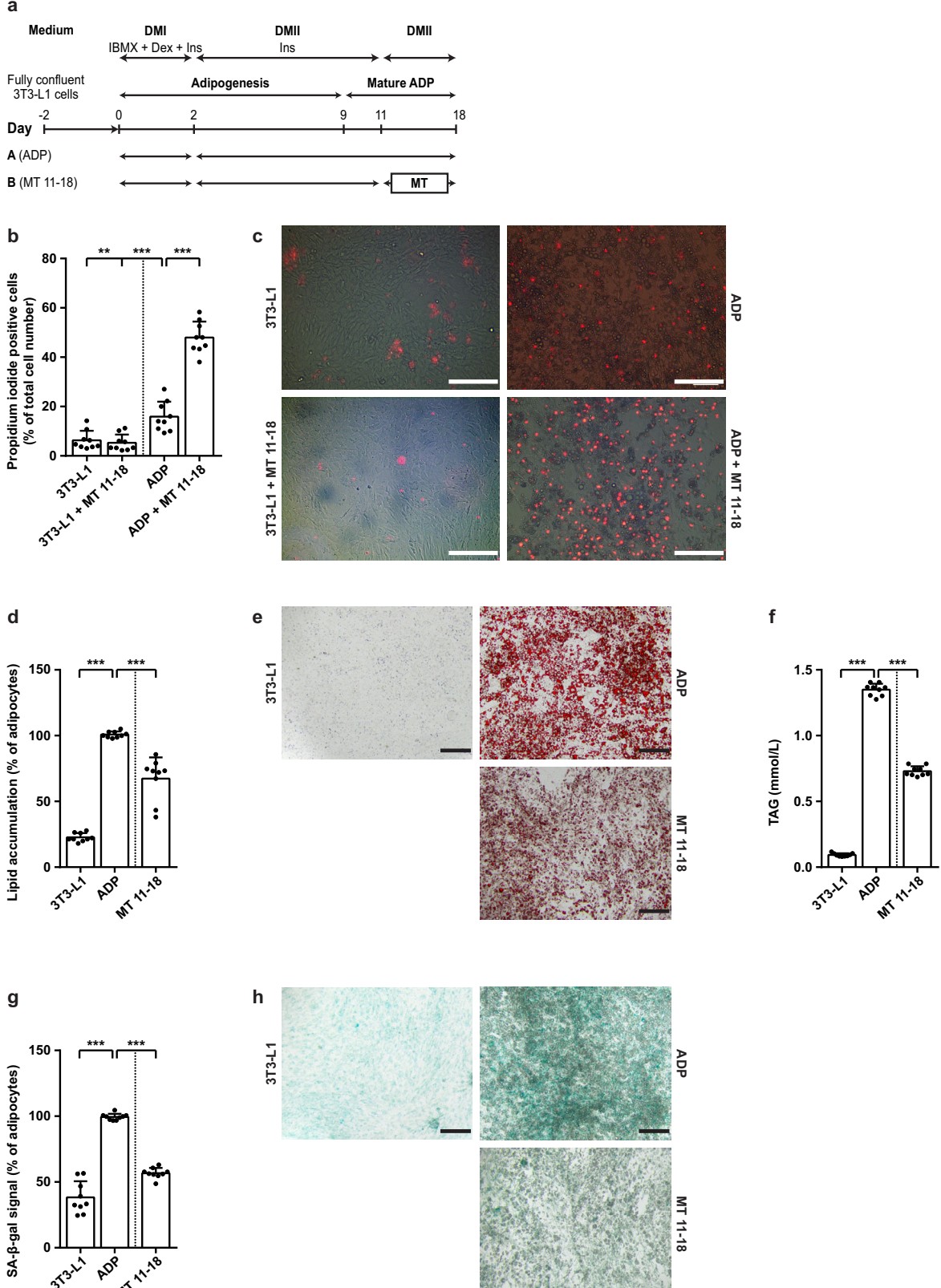

## Methods

**Ethical statement**. This research complies with all relevant ethical regulations. Experiments were performed in agreement with the Animal Protection Law of the Czech Republic and were approved by the Ethics Committee of the Institute for Clinical and Experimental Medicine, Prague (permit number 41/2018) and the Ethics Committee of the Institute of Molecular Genetics, Prague (permit number 51/2018). The human study was approved by the Human Ethics Review Board of

IKEM. All participants signed written informed consent prior to enrollment into the study.

**Cell culture**. Mouse 3T3-L1 pre-adipocytes were purchased from the American Type Culture Collection (USA, Manassas, VA) and cultured in Dulbecco's modified Eagle's medium containing 4.5 g/L D-glucose and 1 mM sodium pyruvate

**Fig. 6 MitoTam eliminates mature adipocytes. a** Time schedule of the 3T3-L1 pre-adipocytes (3T3-L1) differentiated into mature adipocytes (ADP) and then treated with MitoTam (MT, 0.32 μM) or the vehicle at time points as indicated. **b** Quantification of propidium iodide signal accumulated in dead cells was performed ($n = 9$; ADP vs. 3T3-L1 $p = 0.001$; ADP vs. 3T3-L1 + MT 11–18 $p < 0.001$; ADP vs. ADP + MT 11–18 $p < 0.001$); **c** Representative images are shown, the bar indicating 200 μm. **d** Quantification of intracellular lipid accumulation using Oil Red O staining in cells was performed ($n = 9$; ADP vs. 3T3-L1 $p < 0.001$; ADP vs. MT 11–18 $p < 0.001$); **e** Representative images are shown. The bar indicates 200 μm. **f** Level of triglycerides produced into the medium was evaluated ($n = 9$; ADP vs. 3T3-L1 $p < 0.001$; ADP vs. MT 11–18 $p < 0.001$). **g** Quantification of senescent cells was based on SA-β-gal activity ($n = 9$; ADP vs. 3T3-L1 $p < 0.001$; ADP vs. MT 11–18 $p < 0.001$); **h** Representative images are shown, the bar indicating 200 μm. One-way ANOVA, Dunnett's multiple comparisons test was used. The results are derived from at least three independent biological experiments. Data are expressed as mean ± SD; *$p < 0.033$; **$p < 0.002$; ***$p < 0.001$. Source data are provided as a Source Data file.

(Gibco, Carlsbad, CA, USA), and supplemented with 10% fetal bovine serum (Gibco), 4 mM L-glutamine (Gibco), 100 U/mL penicillin, 100 μg/mL streptomycin sulfate (Sigma, St. Louis, MO, USA). Two days after reaching confluence, the cells were differentiated into mature adipocytes by the addition of differentiation medium I (DMI) supplemented with 0.4 μM dexamethasone (Dex; Sigma), 0.5 mM 3-isobutyl-1-methylxanthine (Sigma), 1.7 μM insulin (Ins; Sigma) and 25 mM HEPES (Sigma). Two days later, the cells were switched to DMII supplemented only with 1.7 μM insulin and 25 mM HEPES. In most experiments, MitoTam and tamoxifen were used at 0.32 μM, which is comparable to the concentration of MitoTam detected in adipose tissue in our in vivo experiments. The cells were kept at 37 °C under 5% $CO_2$ in a humidified atmosphere.

### Animal studies

*MitoTam treatment of aged mice.* Aged (18 months) and young (8 weeks) C57BL/6 mice (males + females; randomized in groups; purchased from Charles River Laboratories) were treated either with MitoTam (2 mg/kg of body weight) dissolved in 4% ethanol in corn oil or the vehicle given intraperitoneally (i.p.) once a week for a period of 4 weeks. At the end of the experiment, organs were collected and frozen at −80 °C until further analysis.

*MitoTam treatment of obese and prediabetic mice.* For the experiment, C57BL/6 mice (8 weeks old males; purchased from Velaz, Prague, Czech Republic) were fed an HFD for 6 months. For HFD we used either ssniff® E15742–347 (prolonged effect of MitoTam) or it was composed of 40% of ground rat/mouse maintenance Sniff V1530 (ssniff-Spezialdiäten GmbH, Ferdinand - Gebriel-Weg 16, D-59494, Soest), Sunar Complex 3 (Hero Czech, Prague, Czech Republic) (34%), corn starch (1%), and pork lard (25%). Control mice were fed SD containing rat/mouse maintenance ssniff® V1534–727 (10 mm pellets) (ssniff-Spezialdiäten GmbH). Mice were fed for 6 months to induce obesity and prediabetes, which were assessed by average body weight (46.4 ± 4.9 g for HFD-fed mice, 31.4 ± 1.1 g for SD-fed mice) and fasting glycemia HFD (8.4 ± 0.5 mmol/L for HFD-fed mice, 4.4 ± 1.0 mmol/L for SD-fed mice). The total of 40 mice was divided into four groups, 10 mice each. The animals were fed SD or HFD; at the end of the 6-month period, both groups were treated either with MitoTam (2 mg/kg of body weight) dissolved in 4% ethanol in corn oil or with the vehicle, both administered i.p. twice per week for 4 weeks. Body weight was taken once per week during the 4 weeks of treatment. Food consumption was evaluated once per week, daily FI was estimated as kcal per day during the study. At the end of the experiment, animals were sacrificed between 9 and 11 am by anesthesia: Zoletil 100 at 5 mg/100 g b.w. (Virbac, Carros, France) and Rometar 20 at 2 mg/100 g b.w. (Rometar 20; Bioveta, Ivanovice, Czech Republic), and plasma/serum and organs were collected and frozen at −80 °C until further analysis.

*Comparison of MitoTam and tamoxifen treatment in obese and prediabetic mice.* C57BL/6 mice (16 months old males; purchased from Velaz) were fed HFD (as described above) for 15 months to induce obesity and prediabetes, which were assessed on the basis of body weight of 56.1 ± 5.7 g and fasting glycemia of 7.5 ± 1.4 mmol/L. In all, 21 mice were divided into three groups of six to eight mice each and treated with MitoTam (2 mg/kg of body weight) dissolved in 4% ethanol in corn oil, tamoxifen (2 mg/kg of body weight) dissolved in 4% ethanol in corn oil or vehicle administered i.p. twice per week for 4 weeks. Body weight was taken once per week during the 4 weeks of treatment. Food consumption was evaluated once per week, daily FI was estimated as kcal per day during the study. At the end of the experiment, animals were sacrificed between 9 am and 11 am by anesthesia: Zoletil 100 at 5 mg/100 g b.w. and Rometar 20 at 2 mg/100 g b.w., and plasma/serum and organs were collected and frozen at −80 °C until further analysis.

*Prolonged effect of MitoTam in prediabetic mice.* C57BL/6 male mice (purchased from Charles River Laboratories) were fed either SD or HFD (according to the ssniff® protocol, cat. no. D12492, ssniff-Spezialdiäten, Soest, Germany) for 20 weeks starting ~6 weeks of age. Mice were then treated with MitoTam (2 mg/kg of body weight dissolved in 4% ethanol in corn oil) or with the vehicle given i.p. twice per week for 4 weeks. Body weight was assessed once per week during the course of the experiment. All tests (indirect calorimetry, ECG, IPG, microCT body composition; all described below) were accomplished 3 days or 1 month after the last treatment. After sacrifice, organs were collected, weighed, and frozen at −80 °C until analysis.

*Pair-feeding study in HFD-fed mice.* C57BL/6 J mice (10 months old males; purchased from Velaz) were fed HFD (as described above) for 9 months to induce obesity and prediabetes, and assessed for body weight of 49.9 ± 3.2 g and fasting glycemia of 6.7 ± 0.3 mmol/L. 20 single-housed were divided into 3 groups: group HFD + MT ($n = 7$) was treated with MitoTam (2 mg/kg of body weight) dissolved in 4% ethanol in corn oil, groups HFD + CO PF ($n = 7$) and HFD + CO ($n = 6$) were treated with the vehicle by i.p. administration twice per week for 4 weeks. At the end of the experiment, animals were sacrificed between 9 and 11 am by anesthesia: Zoletil 5 mg/100 g b.w (Zoletil 100; Vibrac) and Rometar 2 mg/100 g b.w. (Rometar 20; Bioveta), and plasma/serum and organs were collected and frozen at −80 °C until further analysis.

Baseline values were taken by evaluating FI every day in single-housed mice for two weeks. With the onset of treatment with MitoTam or vehicle, we started on pair feeding. HFD + MT mice were provided with excess of food in the cage top hopper. HFD + CO PF mice were fed the average amount of food that HFD + MT mice consumed the previous day. 25% of the food amount for the pair-feed group was given to mice at 8 a.m. and the rest of the food (75%) was given at 4 p.m. HFD + CO control mice were provided with excess of food in the cage top hopper.

All mice were maintained at 22 °C, relative humidity (60 ± 10%), and 12 h/12 h light/dark regimen with water and food provided ad libitum, unless stated otherwise. Male mice were used in the experiments containing HFD because of their better hormonal stability and ability to develop obesity and prediabetes.

**microCT body composition.** Body composition was assessed by SkyScan 1176 in vivo microCT (Bruker, Billerica, MA, USA). Mice were anesthetized by intra-muscular injection of 60 μl/20 g mouse body weight of solutions of Zoletil (20 mg/ml) and xylazine (1 mg/ml). The imaging parameters were as follows: pixel size 35 μm, Al filter, no averaging, 180 deg scanning; voltage and current were set to 50 kV and 160 μA, respectively. The reconstruction of projection data was accomplished in InstaRecon (InstaRecon, Champaign, IL, USA) with 3D reconstruction parameters: smoothing, 7, ring artifacts reduction, 5, beam-hardening correction, 10%. The reconstructed VOI for fat tissue was selected according to the X-ray absorption density and quantified using the CtAn software package (Bruker).

**Intraperitoneal glucose tolerance test (IPG).** Mice were fasted overnight for 16 h, weighed, and injected i.p. with 20% glucose to reach 2 g of glucose per kg body weight. Blood glucose was monitored at 0 (basal glucose levels), 15, 30, 60, and 120 min after glucose injection, using the FreeStyle Freedom Lite Glucometer (Abbott Diabetes Care, Alameda, CA, USA).

**Oral glucose tolerance test.** oGTT was performed during the last week of the experiment. Prior to the test, mice fasted for 16 h. Baseline glucose was evaluated in blood samples taken from a tail cut (by removing the distal 2 mm of the tail) using the Accu-chek glucometer (Roche, Basel, Switzerland). The mice then received 2 g/kg body mass of 400 g/L glucose solution (Ardeanutrisol G 40, Ardeapharma, Ševětín, Czech Republic) in sterile water delivered by oral gavage. Glucose levels were assessed at 0, 15, 30, 60, 120, and 180 min after its administration.

**Biochemical assays.** The concentration of triglycerides (TAG) was quantified using a standard enzymatic method assay (TG L 250 S; Erba-Lachema, Brno, Czech Republic). Hormone and cytokine levels in serum were assessed using LUMINEX kits (GIP, insulin, leptin, MMHMAG-44K; MCP1, TNFα, MHSTCMAG-70K; EuroRad, Prague, Czech Republic) and measured on MAGPIX system (Luminex corporate, Austin, TX USA).

For determination of TAG in the liver, the tissue was homogenized and TAG extracted for 16 h with chloroform:methanol (2:1) followed by the addition of 2% $KH_2PO_4$. After 24 h, the organic phase was removed and evaporated. The resulting pellet was dissolved in isopropyl alcohol and TAG content determined using an enzymatic assay (TG L 250 S, Erba-Lachema).

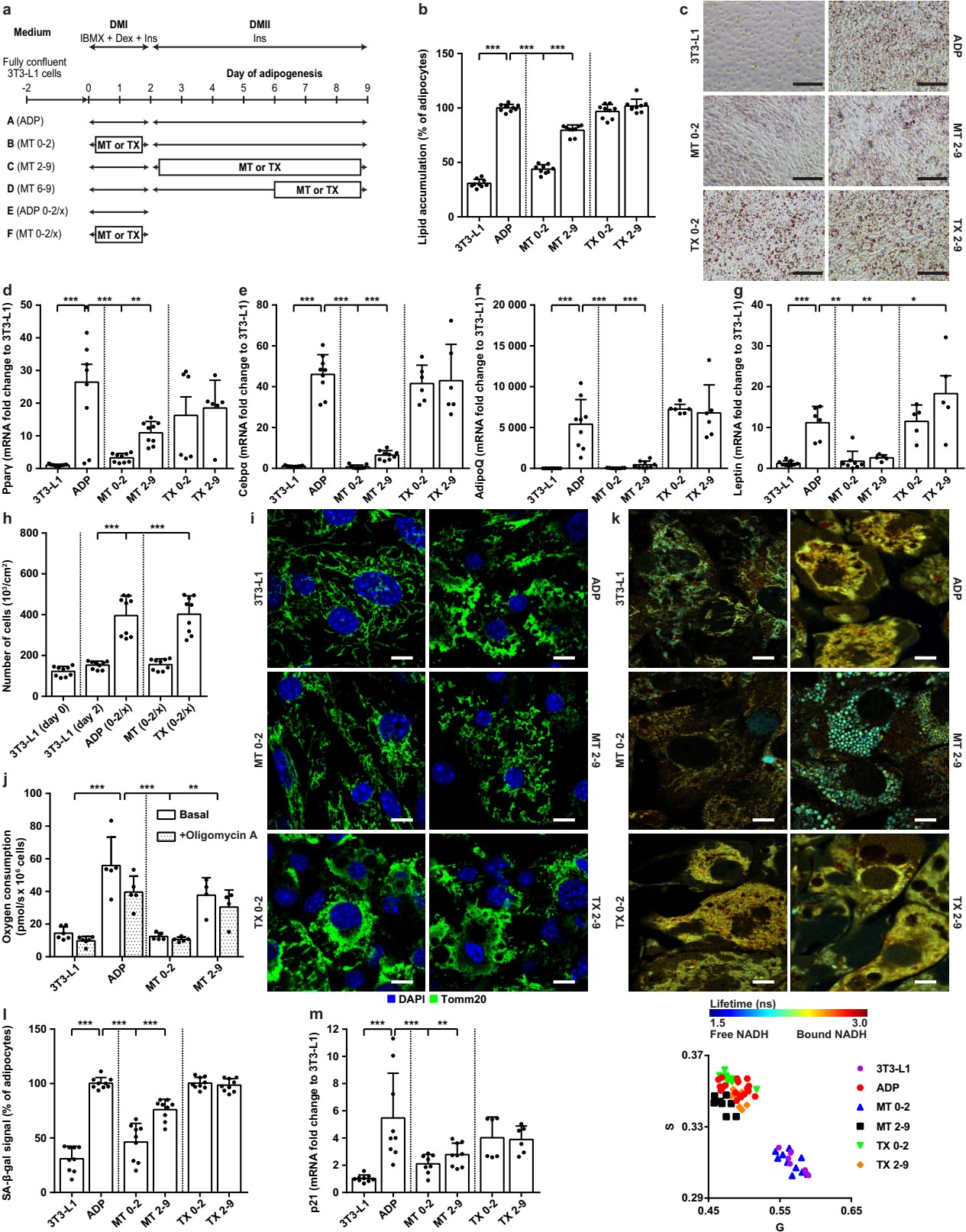

**Quantitative real-time PCR (qRT-PCR).** Small pieces of tissue (1–2 mm$^3$) were placed into 500 µL of RNAzol (or 400 µL for cells seeded in 4 cm$^2$ dishes) (BioRad, Hercules, CA, USA), homogenized (3 × 40 s at 5600 rpm) using the Precellys 24 homogenizer (Bertin Instruments, Montigny-le-Bretonneux, France) and cleared by centrifugation; total RNA was isolated according to the manufacturer's protocol. First-strand cDNA was synthesized from 1 µg of total RNA with random hexamer primers using Revert Aid First-strand cDNA Synthesis Kit (Thermo Fisher). qRT-PCR was performed using the C1000™ Thermal Cycler (BioRad) with 5x HOT

FIREPol® Evagreen® qPCR Supermix (Solis Biodyne, Tartu, Estonia). The relative quantity of cDNA was estimated by the ΔΔCT method, data were normalized to housekeeping genes B2M or β-actin.

**Oil Red O staining.** Mature 3T3-L1 adipocytes were washed once with phosphate-buffered saline (PBS), fixed in 4% solution of formaldehyde in PBS for 30 min, and stained with Oil Red O working solution (0.3% w/v Oil Red O in 60% isopropanol)

**Fig. 7 MitoTam inhibits differentiation of adipocytes.** 3T3-L1 pre-adipocytes were differentiated into mature adipocytes (ADP) while treated with MitoTam (MT, 0.32 μM), tamoxifen (TX, 0.32 μM) or the vehicle at time points as indicated. **a** Time schedule of 3T3-L1 cell exposure to MitoTam or tamoxifen during adipogenesis. **b** Quantification of intracellular lipid accumulation using Oil Red O staining ($n = 9$; ADP vs. 3T3-L1 $p < 0.001$; ADP vs. MT 0–2 $p < 0.001$; ADP vs. MT 2–9 $p < 0.001$); **c** representative images are shown, the bar indicating 100 μm. Transcripts of **d** $Ppar\gamma$ (3T3-L1, ADP, MT 0–2, MT 2–9 $n = 9$; TX 0–2, TX 2–9 $n = 6$; ADP vs. 3T3-L1 $p < 0.001$; ADP vs. MT 0–2 $p < 0.001$; ADP vs. MT 2–9 $p = 0.004$); **e** $Cebp\alpha$ (3T3-L1, ADP, MT 0–2, MT 2–9 $n = 9$; TX 0–2, TX 2–9 $n = 6$; ADP vs. 3T3-L1 $p < 0.001$; ADP vs. MT 0–2 $p < 0.001$; ADP vs. MT 2–9 $p < 0.001$); **f** $AdipoQ$ (3T3-L1, ADP, MT 0–2, MT 2–9 $n = 9$; TX 0–2, TX 2–9 $n = 6$; ADP vs. 3T3-L1 $p < 0.001$; ADP vs. MT 0–2 $p < 0.001$; ADP vs. MT 2–9 $p < 0.001$) and **g** leptin (3T3-L1 $n = 9$; ADP $n = 6$; MT 0–2 $n = 8$; MT 2–9, TX 0–2, TX 2–9 $n = 5$; ADP vs. 3T3-L1 $p < 0.001$; ADP vs. MT 0–2 $p = 0.001$; ADP vs. MT 2–9 $p = 0.01$; ADP vs. TX 2–9 $p = 0.04$) were quantified by qRT-PCR. **h** Mitotic clonal expansion shown as number of 3T3-L1 cells after the first phase of differentiation (0–2/x) ($n = 9$; 3T3-L1 (day 2) vs. ADP (0–2/x) $p < 0.001$; 3T3-L1 (day 2) vs. TX (0–2/x) $p < 0.001$). **i** Mitochondrial morphology documented by Tomm20 immunofluorescent staining, DAPI denoting cell nuclei. The bar indicates 10 μm. **j** Basal and uncoupled (+oligomycin) cellular respiration of pre-adipocytes in the absence or presence of MitoTam was evaluated (3T3-L1 $n = 6$; ADP, MT 0–2 $n = 5$; MT 2–9 $n = 4$; ADP vs. 3T3-L1 $p < 0.001$; ADP vs. MT 0–2 $p < 0.001$; ADP vs. MT 2–9 $p = 0.01$). **k** Representative images of mitochondrial NAD(P)H signal measured by two-photon FLIM microscopy. Fluorescence lifetime is color-coded as shown. The bar indicates 10 μm (top). Dots in the phasor display (bottom) represent average values from individual fields of view (3T3-L1 $n = 8$; ADP $n = 19$; MT 0–2 $n = 10$; MT 2–9, TX 0–2 $n = 8$; TX 2–9 $n = 9$). G and S values correspond to real and imaginary parts of first components of Fourier transformation. **l** Quantification of senescent cells assessed by SA-β-gal activity ($n = 9$; ADP vs. 3T3-L1 $p < 0.001$; ADP vs. MT 0–2 $p < 0.001$; ADP vs. MT 2–9 $p < 0.001$); and **m** The level of $p21^{waf1}$ assessed by qRT-PCR (3T3-L1, ADP $n = 9$; MT 0–2 $n = 8$; MT 2–9 $n = 9$; TX 0–2, TX 2–9 $n = 6$; ADP vs. 3T3-L1 $p < 0.001$; ADP vs. MT 0–2 $p < 0.001$; ADP vs. MT 2–9 $p = 0.006$). **b, d–h, l, m** One-way ANOVA, Dunnett's multiple comparisons test was used. **j** Two-way ANOVA, Dunnett's multiple comparisons test was used. The results are derived from at least two or three independent biological experiments. Data in all experiments are expressed as mean ± SD; *$p < 0.033$; **$p < 0.002$; ***$p < 0.001$. Source data are provided as a Source Data file.

for 40 min at room temperature (RT). The cells were then washed five times with distilled water. The signal was detected by light microscopy (Leica Microsystems, Mannheim, Germany). For quantification, Oil Red O was extracted into 100% isopropanol for 10 min and absorbance read at 500 nm.

Frozen liver tissue was cut using the microtome into 10-μm slices and fixed in 4% formaldehyde in PBS for 10 min at RT. The tissue was washed with water and distilled water, and rinsed in 60% isopropanol for 5 min. Lipids in the tissue were stained with Oil Red O working solution (see above) for 20 min at RT and washed in 60% isopropanol for 5 min and distilled water for 5 min. The nuclei were stained with hematoxylin for 5 min, and the tissue was washed twice with distilled water for 5 min each. Coverslips were mounted in Mowiol and the signal detected by light microscopy (Leica Microsystems) and evaluated using the Fiji software.

**Evaluation of respiration.** The high-resolution Oxygraph-2k respirometer (Oroboros Instruments, Innsbruck, Austria) was used to assess routine and uncoupled respiration. Cells were trypsinized, washed with PBS, re-suspended at $2 \times 10^6$ cells per mL in the Mir05 medium (0.5 mM EGTA, 3 mM MgCl$_2$, 60 mM κ-lactobionate, 20 mM taurine, 10 mM KH$_2$PO$_4$, 110 mM sucrose, 1 g/L essentially fatty acid-free bovine serum albumin, 20 mM Hepes, pH 7.1 at 30 °C) and transferred to the chamber of the Oxygraph-2k instrument. Respiration evaluation was performed at 37 °C. After signal stabilization, the chamber was closed and the oxygen consumption started to rise to show the level of basal respiration. To measure uncoupled respiration, oligomycin A (2 μg/ml) was added. Data were evaluated using DatLab5 software (Oroboros Instruments).

**Evaluation of mitochondrial membrane potential.** To assess $\Delta\Psi_{m,i}$, cells were seeded in 24-well plates and treated with MitoTam as described above. Tetramethylrhodamine methyl ester (TMRM; Sigma) at 50 nM was added to all wells and the cells were incubated for 30 min prior to analysis by flow cytometry using the LSR Fortessa instrument (Beckton Dickinson, Franklin Lakes, NJ, USA). Prior to analysis, control cells were also pre-treated with 1 μM of carbonyl cyanide 3-chlorophenylhydrazone (Sigma) for 10 min. Live cells were gated based on FCS and SSC. After single cells were gated (FCS-A to FSC-H), they were analyzed for the TMRM signal in the PE-Texas Red channel.

**Assessment of cell death.** The medium containing dead cells was collected into a clear tube, adherent cells were trypsinized, re-suspended in the medium containing dead cells, and centrifuged at $500 \times g$ for 5 min. The pellet was re-suspended in 200 μL of annexin V-binding buffer containing 0.3 μL of annexin V-Dyomics 647 (Apronex, Prague, Czech Republic), and incubated for 20 min at 4 °C. Hoechst 33258 (5 μg/ml) was added before the cells were analyzed using the LSR Fortessa flow cytometer. Cell death was expressed as the percentage of annexin V-positive/Hoechst-positive cells.

Alternatively, 3T3-L1 cells were seeded in 12-well plates, differentiated into mature adipocytes, and treated with MitoTam as described above. After treatment, propidium iodide at 1 μg/ml was added to the medium, and cells were observed by fluorescence microscope (Leica Microsystems) and evaluated using the ImageJ 1.52 v software (National Institutes of Health, Bethesda, MD, USA). Cell death was expressed as the percentage of propidium iodide-positive cells of the total cell number. Live cells were gated based on FCS and SSC. After single cells were gated

(FCS-A to FSC-H), they were analyzed for death in the Indo-1 (violet) channel (Hoechst) and the APC-A channel (annexin V).

**Indirect immunofluorescence.** Cells grown on glass coverslips were fixed with 4% paraformaldehyde (PFA; VWR, Radnor, PA, USA) and permeabilized with 0.1% Triton X-100 (Sigma) in two consecutive steps, each at RT for 15 min. After washing with PBS, cells were incubated in 10% FBS (diluted in PBS) for 30 min to block unspecific signals. Cells were then incubated with diluted primary antibodies (1 h; RT), washed extensively with PBS/0.1% Tween 20, and incubated with secondary antibodies (1 h; RT).

Tissue sections were permeabilized for 15 min in dimethyl sulfoxide (DMSO, Sigma) and for 30 min in Saponin (ThermoFisher) followed by blocking in 10% FBS (diluted in PBS) for 1 h. Each step was preceded by two 10 min washes with PBS. After blocking, the sections were incubated under vacuum with the diluted primary antibody (overnight, RT), washed three times for 5 min each with PBS, and incubated in the dark with the secondary antibody (1 h, RT). Coverslips were mounted in Mowiol containing 4',6-diamidino-2-phenylindole (Sigma) to stain nuclei, and the signal was detected using the Leica SP8 FLIM confocal microscope (Leica Microsystems).

**Detection of senescence-associated β-galactosidase activity.** The tissue was cut into small pieces (1–2 mm$^3$) and fixed in 1% PFA/0.2% glutaraldehyde at 4 °C for 1 h. To detect SA-β-gal activity the tissue was washed with PBS (pH 5.5), supplemented with 1 mM MgCl$_2$ and stained with the X-gal solution (1 mg/mL X-gal, 0.12 mM K$_3$Fe(CN)$_6$, 0.12 mM K$_4$Fe(CN)$_6$, 1 mM MgCl$_2$ in PBS at pH 5.5) at 37 °C for 3–5 h. To stop the reaction, tissue was washed three times with water, and the blue signal was detected using a light microscope (Leica, Mannheim, Germany) and evaluated using the Photoshop and ImageJ program as an average of five sections per sample.

Mature 3T3-L1 adipocytes were washed twice with PBS, fixed with 0.5% glutaraldehyde in PBS for 10 min and twice with PBS (pH 5.5) supplemented with 1 mM MgCl$_2$. The cells were stained with the X-gal solution (see above) at 37 °C for 5 h in the dark. After washing three times with PBS, SA-β-gal signal was detected by light microscopy (Leica Microsystems or Olympus IX83) and evaluated using the ImageJ 1.52v software.

**Tissue processing.** For paraffin embedding, tissue was fixed overnight in 2% PFA, placed in 70% EtOH, and processed using the Leica ASP6025 autoprocessor. The program was set to 12 h standard dehydration and paraffin saturation protocol. After processing, tissue samples were embedded into formalin-fixed paraffin blocks upside down by cutting their surface and orienting the blocks transversally. Standard hematoxylin and eosin staining, dehydration, and placing the samples on coverslips were accomplished using the Leica ST5020 automated staining instrument in combination with the Leica CV5030 cover-slipper.

**Two-photon microscopy.** NADH was assessed by two-photon microscopy using the Zeiss LSM880 NLO inverted microscope equipped with the Chameleon Ultra II and Compact OPO MP lasers (Coherent, Santa Clara, CA, USA), internal 32 channel spectral GaAsP detector, and two non-descanned BiG-2 GaAsP detectors. The ×63/1.4 Plan-Apochromat oil immersion objective, 8 mW of 740 nm laser line, and emission range 390–480 nm were used. NADH emission spectrum was first

confirmed with the internal spectral detector; for the final image acquisition, we used the non-descanned BiG detector. The cells were grown in glass-bottom microscopy plates (D35-14-1.5-n, Cellvis, Mountain View, CA, USA) in a standard culture medium. Mitochondrial NADH was quantified by Fiji by means of segmenting cells using thresholding and averaging the signal over the mitochondrial regions of interest.

**Statistical analysis.** Unless stated otherwise, data are mean values ± standard error of means (SEM) of at least three independent experiments. In mouse experiments, groups of 6–10 animals were used, unless stated otherwise. One-way ANOVA and two-way ANOVA (for body weight and FI data) followed by Tukey's multiple comparisons test or unpaired $t$ test presented as mean ± SEM were used to assess statistical significance with $p < 0.033$ being regarded as significant, using the GraphPad Prism software. Images are representative of at least three independent experiments.

**Reporting summary.** Further information on research design is available in the Nature Research Reporting Summary linked to this article.

## Data availability

All the relevant data supporting the findings are available within this article or in the Supplementary Information file. The data sets generated and/or analyzed during the current study are available from the corresponding authors on a reasonable request. A reporting summary for this article is available as a Supplementary Information file. The mass spectrometry proteomics data have been deposited to the ProteomeXchange Consortium via the PRIDE[56] partner repository with the dataset identifier PXD031710 and https://doi.org/10.6019/PXD031710. Source data are provided with this paper.

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

## Acknowledgements

This study was supported by grants from the Czech Science Foundation (18-02550 S to S.S.H, and 18–10832 S, 20-05942 S, and 21-04607X to J.N.), the Czech Health Science Foundation (17-30138 A and 21-03-00545 to J.N.), the BIOCEV European Regional Development Fund CZ.1.05/1.100/02.0109, the Institute of Biotechnology (RVO: 86652036) and by CZ—DRO ("Institute for Clinical and Experimental Medicine—IKEM, IN 00023001") to M.H. E.V. was supported by the Grant Agency of Charles University (GA UK 1560218). P.S. was supported by UCT Prague Rector´s Junior Grant for the year 2020. B.E. was supported by the International Mobility Program funded by MEYS CR. We acknowledge the Imaging Methods Core Facility at BIOCEV, an institution supported by the Czech-BioImaging large RI projects (LM2015062 and CZ.02.1.01/0.0/0.0/16_013/0001775, funded by MEYS CR) for their support with obtaining imaging data presented in this paper. We acknowledge Karel Harant and Pavel Talacko from Laboratory of Mass Spectrometry, Biocev, Charles University, Faculty of Science for performing LC/MS analysis.

## Author contributions

E.V., J.T., P.S., V.S., V.N., D.P.R., S.P., B.P., F.S., D.Z., M.M., J.P., B.E., R.S., M.H., and S.S.H. performed experiments and analyzed data, J.S. and L.W. synthetized MitoTam, Z.B. provided clinical data, E.V., J.T., P.S., J.P., S.S.H., M.H., and J.N. designed experiments, and wrote and edited the manuscript. All authors reviewed and edited the manuscript.

## Competing interests

S.S.H., J.N., J.S., and L.W. are co-inventors of MitoTam as a senolytic agent, and J.N., J.S., and L.W. are co-owners of MitoTax s.r.o. and the MitoTam intellectual property, together with KKCG a.s. and SmartBrain s.r.o., who financed the MitoTam-01 oncolytic trial. Other authors declare no conflict of interest.
