## [Peer Review File · Nature Communications]

Reviewers' comments:

Reviewer #1 (Remarks to the Author):

The manuscript by Vacurova and colleagues proposes that mitochondrially targeted tamoxifen has senolytic activity in vivo and can improve specific health parameters in a mouse model of obesity. I found some of the observations to be of great interest and potentially important- however, there are some significant flaws which limited my enthusiasm.

I believe that the manuscript would benefit from a more detailed explanation of the mechanism of action of mitoTAM- why does it act as a senolytic? What specific survival mechanisms of senescent cells is mitoTAM targeting?

Furthermore, it is not clear why authors have compared tamoxifen with mitoTAM- is there any evidence that Tamoxifen alone can act as senolytic? What is the rationale for this experiment?

Finally, the authors show in vitro evidence that MitoTAM suppresses differentiation of pre-adipocytes. While it is an interesting observation- there is no sufficient explanation as to why this was evaluated, why it is important and importantly what is the mechanism of action.

Specific comments:

Figure 1- The authors use different animal numbers per group throughout this figure, but stated to have used 5 as control and 6 for treatment. If animals are included/excluded it should be stated why and how it was determined.

Evidence for efficacy of mitoTAM as a senolytic in vivo during aging is limited. Characterization of senescent markers lacks consistency with some markers being measured in 1 organ but not another. No SASP factors were evaluated.

Authors claim that mitoTAM “completely reversed” certain parameters of glucose metabolism- however, this is not supported by the data.

Figure 1a, b – Images should be provided for the SA-beta-Gal staining. Methods section needs updating as it is not understandable how the data was generated. Methods should be written in detail and not referred to previous publications.

Figure 1c-e - Why are not all 3 senescence markers tested in spleen, liver and lungs?

SFigure 1a – The image of SA-beta-Gal is not acceptable. Not sure what I am supposed to see here.

SFigure 1b – Why is now PAI measured? Not assessed in aged animals (F1).

Figure 2- It concerns me that mitoTAM leads to a progressive decrease in body weight and it was not clear to me why this occur- is it due to decreased food consumption? It is interesting that other published senolytic agents did not result in decreased body weight in high fat diet- how do the authors explain this?

No error bars in SD controls (Figure 2a)

SFigure 2h – CD14 is not significantly increased and treatment didn't reverse immune infiltration.

Figure 2h – Lipid accumulation should be analysed and not only H&E images. Authors should include Nile Red, Oil Red O or Bodipy staining.

SFigure 2k – Higher magnification images are needed and staining should be analysed.

Figure 3- Not clear in the text the rationale behind comparison of mitoTAM and Tamoxifen.

Figure 3h – Lipid accumulation should be analysed and not only H&E images. Authors should include Nile Red, Oil Red O or Bodipy staining.

Figure 4f – Not sure what I am supposed to see here. This needs more explanation and analysis.

Reviewer #2 (Remarks to the Author):

General:

The use of mitochondrially targeted tamoxifen (MitoTam, MT) for the treatment of obesity and type 2 diabetes via elimination of senescent cells is an interesting and quite novel approach. The data presented are thus of considerable interest although largely descriptive. However, a somewhat deeper scrutiny of possible mechanisms would be desirable.

Given the fact that MT affects mitochondrial respiration by its interaction with respiratory complex 1, this immediately brings to mind the concept of mitohormesis which is used to explain the beneficial systemic effects of mitochondrial stress, and has attracted a lot of attention recently (for reviews see: Bar Ziv et al. 2020, doi: 10.15252/embr.202050094; Conte et al. 2020, doi: doi: 10.1007/s00281-020-00813-0; Klaus & Ost 2020, doi: 10.1016/j.exger.2019.110796). The systemic effects of MT are very similar to effects of the stress induced mitokines FGF21 and GDF15 (see also my specific comment below), so I think this is an important aspect which should be addressed.

Specific remarks:

1. The title might be a bit too strong as a statement. There was no real diabetes model investigated. C57BL/6N mice used here do become obese and insulin resistant when fed a high fat diet, but they never become really diabetic. Additional, maybe genetic diabetes models would need to be studied to draw that conclusion.

2. Methods: The sex of the animals is not specified for all experiments. Considering that tamoxifen is a selective estrogen receptor modulator, it would be important to use both male and female mice (also considering possible side effects).

3. Results: The statement in lines 113-116 does not quite match the corresponding figure (1F,G). It is stated that aged mice showed an increased fasting glucose and poor glucose clearance. However, as presented in the figure these were not different from young CO. Aged MT treated mice rather showed very low glucose levels during the oGTT, which were even lower than in young-CO. To me this looks like an impairment of intestinal glucose uptake in the Aged+MT group rather than an improved glucose uptake. It would be helpful to also show insulin data during the oGTT. Also, data on food intake and body weight development during the MT treatment of aged mice would be of importance.

4. Results: There is a discrepancy between the data on food intake presented in Suppl. Fig. 2B (no effect of MT on food intake on either HFD or LFD) and suppl Fig 4b (considerably reduced food intake by MT on HFD one month after treatment). Could this be due to long term side effects of MT?

5. Regarding the comparison between MT and TX treatment: in both treatments the same concentration (2mg/kg BW) was used. What was the rationale for using this concentration?

6. There is no suggestion or discussion about the specific mechanisms of MT compared to TX. It is known that mitochondrial stress leads to an activation of the integrated stress response via ATF4 resulting in increased expression of mitokines such as FGF21 and GDF15, which both have been shown to improve diabetes related features. It would be interesting to measure their plasma levels and gene expression e.g. in liver, muscle or adipose tissue. In this respect it would also be of interest to look for possible browning of subcutaneous adipose tissue (e.g. UCP1 gene expression).

7. Regarding the strong effect of MT on body fat reduction in HFD fed mice (Fig. 2) I am wondering about possible effects on intestinal energy absorption (which could also explain the decreased glucose levels during oGTT after MT treatment in aged mice, Fig. 1). Since enterocytes have a very high metabolic activity, disturbance of their mitochondrial respiratory activity might have considerable effects on their activity relating to nutrient absorption. This could easily be addressed in a first step by measuring food assimilation, i.e. energy intake minus fecal energy excretion (measured by bomb calorimetry).

8. The considerable loss of body weight (Fig. 2A) of over 10 grams in 4 weeks is much higher than the loss of VAT (about 1 g). Even accounting for other fat depots this could also reflect a loss of lean body mass or possibly dehydration. Are there any data available on muscle mass or total lean mass? Was water intake affected by the MT treatment?

9. I do not understand Fig. 4D and the conclusion drawn. In the text it is stated (lines 212-213): "However, change of diet after the end of treatment (HFD to SD) stabilized the improved glucose uptake." If the labeling corresponds to the labeling of the other subfigures, the HFD+MT group is actually missing, only the data for the SD+MT group are shown. Also, I am wondering why for Fig. 4A an IP glucose tolerance test was performed and for Fig. 4D an oral GTT (similar to Fig 2)? Actually, it would be quite interesting to directly compare the results of an oGTT with that of an IP-GTT. If MT indeed affects intestinal nutrient (glucose?) uptake, the results could be quite different. Possible incretin effects would also be bypassed when using an IP-GTT.

10. Regarding the in vitro studies on 3T3-L1 adipocytes: How comparable are the in vitro and in vivo concentrations of MT and TX used?

11. Discussion: The studies using 3T3-L1 adipocytes show nicely that MT directly affects adipocyte differentiation but seemingly has no or little effects on differentiated adipocytes. In my opinion, this does not quite support the conclusion that MitoTam reduces visceral adipose tissue by suppressing adipogenesis since no adipogenesis is necessary for initial adipocyte hypertrophy. Furthermore, adipogenesis does not really play a role in the weight reduction effects in already obese mice. Most importantly, TX basically showed the same fat reducing effect as MT in vivo without affecting adipogenesis in vitro, suggesting other mechanism must be involved.

12. The fact that TX affected body mass and fat in the same way as MT but did not decrease insulin

levels or B-gal positive cells (Fig. 3), suggests that the improvement of glucose metabolism is mainly due to the senolytic effect of MT rather than its effect on adipose tissue mass. This is a highly interesting aspect which should be addressed in more detail. It also makes me wonder why a major focus was put on adipose tissue and adipogenesis. Wouldn't it be more appropriate to focus more on liver or beta-cell function? Also, muscle might be important considering its prominent role in glucose uptake and contribution to overall energy metabolism.

Reviewer #3 (Remarks to the Author):

In this manuscript, Vacurova et al explore the impact of mitochondrially targeted tamoxifen (MitoTam) in a variety of metabolically compromised mouse models. They report that MitoTam improves a number of metabolic parameters and conclude that this is primarily due to effects to reduce senescent cell numbers and to impede adipogenesis. While there is some interesting data in this manuscript, there are a number of significant issues that require attention.

There are a substantial number of instances throughout the manuscript where non significant changes are referred to in a manner that would suggest they are different (both mouse data and human data). The data are either significant or they are not. It is inappropriate to make conclusions based on trends (often weak) and the authors should remove such statements from the manuscript.

The assertion that senescent cells are responsible for many of the metabolic defects in the various mouse models and hence the reduction of senescent cells by MitoTam is a major mechanism underpinning its effects is not especially well supported by the data. For example, in Figure 1 there are substantial differences in senescence markers between the young and old control groups, but the glucose tolerance test between these groups is not statistically different. On the other hand the aged group treated with MitoTam has a very pronounced improvement in glucose tolerance, but shows predominantly non significant changes in senescence markers, particularly in important metabolic tissues such as the liver. Likewise in Figure 3, the tamoxifen group exhibits substantially reduced adiposity and improved glucose tolerance compared to the control group and yet the senescence markers are not different between these groups. These data do not strongly support a causative role for senescent cells in the induction of metabolic defects and thus call into question one of the main mechanisms proposed to underpin the effects of MitoTam. The authors need to address these inconsistencies.

The authors report that there is no change in energy expenditure and no change in food intake – so how is it that the treated mice are losing weight (and fat mass)? The issue is that food intake in this manuscript has been reported normalised to body weight, which largely obscures the fact that the animals treated with MitoTam must be eating less total calories (as they are losing body weight). Energy expenditure has not been normalised to body weight and thus there is an imbalance in these measures. The authors are referred to an excellent review article on assessment of mouse energy metabolism (Nat Methods. 2011, 9: 57–63) where the recommendation is that the analysis of energy balance should be made on the raw data (e.g. calorie intake) expressed per mouse (rather than normalising one side of the

equation by body weight and not the other). It is likely that much of the benefit of MitoTam is in fact a result of an acute drop in caloric intake, which is known to rapidly improve metabolic parameters in mice. A pair feeding study would help to resolve this.

Following on from the above, the authors provide evidence in in vitro experiments that MitoTam influences adipogenesis and propose this might be responsible for the weight (and fat) loss phenotype observed. The concern with this proposal is that the authors have used a model of 'reversing' diet induced defects – namely mice have been fed a high fat diet long term and are then exposed to MitoTam for a short 4 week period. The authors own data (Figure 2A) shows that body weight has stabilised with the long term fat feeding regime and thus under these conditions there is unlikely to be extensive adipogenesis occurring. Hence the proposal that MitoTam is blocking adipogenesis to cause the observed weight loss over the 4 week treatment period is difficult to reconcile. Please comment on this discrepancy and how it impacts on the conclusions put forward in the discussion.

It is hard to gauge from the manuscript which tissues might be responsible for the benefits of MitoTam on glucose tolerance and insulin sensitivity, as many tissues have been assessed for different parameters (i.e. lipid accumulation, senescence markers). The authors should consider conducting hyperinsulinemic/euglycemic clamps to determine if specific tissues (e.g. liver, adipose) may have a more major role in driving the beneficial effects on insulin sensitivity.

Minor points

The statistical analysis section indicates one way ANOVA was used for all mouse studies. For the data in Figure 2, a 2 way ANOVA is likely more appropriate, with diet and treatment as the 2 main factors.

The authors may consider calculating the incremental area under the curve for the glucose tolerance tests rather than the total area under the curve – it might reveal more substantial differences for some of the experiments.

The authors frequently present histological images to highlight lipid accumulation and then complement this with the more robust biochemical assessment of triglyceride levels. They are encouraged to provide a triglyceride measurement for the samples depicted in Figure 4I, to provide continuity for the manuscript and greater confidence in the reported finding.

Line 62 in the introduction implies that ectopic lipid accumulation in non-adipose tissues only occurs once fat storage is exhausted. This is not accurate, as lipid accumulation and insulin resistance occur in peripheral tissues very rapidly on exposure to excess dietary fat (i.e. within days). Please correct this statement.

Line 130 – it is suggested that the fat-fed C57BL/6 mouse model is a model of diabetes, but in fact this mouse strain will become obese and insulin resistance, but never fully develops diabetes. Please correct.

The relevance of Figure 4D is questionable, as the impact of changing from HFD to SD appears to be the overriding factor in these effects, not previous treatment with MitoTam.

Line 335 – the authors indicate that tamoxifen alone had no effect on lipid accumulation, but this is incorrect as per the data shown in Figure 3B and extended Figures 3B and 3G.

Reviewer #1:

The manuscript by Vacurova and colleagues proposes that mitochondrially targeted tamoxifen has senolytic activity in vivo and can improve specific health parameters in a mouse model of obesity. I found some of the observations to be of great interest and potentially important- however, there are some significant flaws which limited my enthusiasm.

We thank the reviewer for considering our work of great interest, and understand that responding to the points raised by him/her is of importance to bring the manuscript to high standard in order that it is publishable in Nature Communication.

I believe that the manuscript would benefit from a more detailed explanation of the mechanism of action of mitoTAM- why does it act as a senolytic? What specific survival mechanisms of senescent cells is mitoTAM targeting?

We agree that this point deserves more focus in order to explain the senolytic activity of MitoTam. In fact, we based our project, in which we studied the effect of MitoTam on T2DM, on the following paper: Hubackova S et al (2019) Selective elimination of senescent cells by mitochondrial targeting is regulated by ANT2. *Cell Death Differ* 26, 276-290 (reference #18 in the manuscript). Here we show that MitoTam limits OXPHOS, which is a situation that causes death to senescent cells, the reason being that the level of ANT2 is low in senescent cells, whereby these cells cannot pump ATP inside mitochondria in order to maintain mitochondrial potential on the bases of cleavage of ATP by ATPase. This results in the collapse of mitochondrial potential and subsequent loss of mitochondrial integrity. Now we discuss this more in detail in the updated version of the manuscript to make it clearer for readers.

Furthermore, it is not clear why authors have compared tamoxifen with mitoTAM- is there any evidence that Tamoxifen alone can act as senolytic? What is the rationale for this experiment?

MitoTam is derived from tamoxifen by tagging the parental compound with the delocalised lipophilic cationic group triphenylphosphonium (TPP) that anchors it at the interphase of the inner mitochondrial membrane and the matrix. We compared the effect of tamoxifen with MitoTam to see whether tamoxifen, as a functional group of MitoTam, is alone responsible for the observed effects or whether it is rather an effect of its specific targeting into mitochondria, since most targets of “free” tamoxifen are present in the cytoplasm.

Finally, the authors show in vitro evidence that MitoTAM suppresses differentiation of pre-adipocytes. While it is an interesting observation- there is no sufficient explanation as to why this was evaluated, why it is important and importantly what is the mechanism of action.

We evaluated the effect of MitoTam on adipogenesis, since we observed that treatment of our diabetic mice with MitoTam reduced their body weight, adipose tissue mass and the deposition of lipids in various tissues. We document the effect of MitoTam on several transcription factors that control adipogenesis. Since most of the factors influencing progression of T2DM are tightly connected with adipose tissue, it is important to study the effect of MitoTam on this tissue. We made this clearer in the updated version of the manuscript.

Specific comments:

Figure 1- The authors use different animal numbers per group throughout this figure, but stated to have used 5 as control and 6 for treatment. If animals are included/excluded it should be stated why and how it was determined.

In Figure 1 we used 5 mice as controls since this was sufficient for statistics purposes, and 7 or 6 mice for aged control and MitoTam treatment, respectively, in order to get more consistent data. In case of lungs, we unfortunately lost one young sample, therefore we have just 4 animals included in the analysis. We are aware of this inconsistency. Therefore, we added new cohort of aged mice in the updated version of the manuscript to Figure 1 to increase the number of mice per group and to make our results are statistically significant. Non-significant data that, nevertheless, support our results, are now in Supplementary Figure 1a.

Evidence for efficacy of mitoTAM as a senolytic in vivo during aging is limited. Characterization of senescent markers lacks consistency with some markers being measured in 1 organ but not another. No SASP factors were evaluated.

As mentioned above, we show the efficacy of MitoTam as a senolytic agent in our previous manuscript (Hubackova S et al. Cell Death Differ 2019, 26, 276-290; reference #18 in our manuscript). Notwithstanding this, we provided more detailed explanation of MitoTam's mechanism of action in the updated version of the manuscript to make it clearer for the reader.

Despite extensive research, there is no specific marker of senescence. Cellular senescence is usually detected using combination of several markers. These markers are expressed differently in different organs. Therefore, analysis of senescent markers appears inconsistent across the tested organs in our manuscript. We tested all markers in all organs, but some of these markers were under the detection limit in one organ and highly expressed in another. We made this clear in the text of updated version of the manuscript and in Figure legends. However, in Figure 1 we now show p21 expression in lungs to make our data more consistent.

Evaluation of SASPs is an excellent point and we thank the Reviewer for it. We analyzed IL6, IL8, MCP1 and TNF α in different tissues of aged mice, but IL6 was the only one increased in aged animals. Although the attached pictures show their decrease after MitoTam treatment, these results are not significant. Therefore, we decided not to include these results in the updated version of the manuscript.

Authors claim that mitoTAM “completely reversed” certain parameters of glucose metabolism- however, this is not supported by the data.

We thank the Reviewer for this comment. We agree that the wording ‘completely reversed’ is too strong and therefore we changed it to ‘reversed’ in the updated version of the manuscript. However, most of our parameters increased in HFD-fed mice were reduced to the level of SD-fed mice (see for example Fig. 2e or 2f), which is consistent with our statement.

Figure 1a, b – Images should be provided for the SA-beta-Gal staining. Methods section needs updating as it is not understandable how the data was generated. Methods should be written in detail and not referred to previous publications.

We agree with the Reviewer and have re-written the Methods section to be more understandable in the updated version of the manuscript.

Figure 1c-e - Why are not all 3 senescence markers tested in spleen, liver and lungs?

As mentioned above, expression of senescent markers is organ-specific. We tested all three senescent markers in all organs and, for example, p19 was not detectable in spleen or liver, but was expressed in lungs. We discuss the different expression of individual senescence markers in the updated version of the manuscript to make clear for readers why we do not show all senescence markers in all organs tested. However, as mentioned above we now show significant changes in p21 expression in lungs in Figure 1 to make our data more consistent. Expression of other genes supporting our hypothesis is now shown in Supplementary Figure 1a.

SFigure 1a – The image of SA-beta-Gal is not acceptable. Not sure what I am supposed to see here.

The image shown in Supplementary Figure 1a (S1b in the updated version of the manuscript) represents a liver slide from control and HFD-fed mice stained for β -gal. The percentage of β -gal positive tissue was calculated using the Photoshop software. Notwithstanding this, we agree that this image can be perceived as unclear and have decided to remove it from the updated version of manuscript.

SFigure 1b – Why is now PAI measured? Not assessed in aged animals (F1).

While increased expression of PAI was detected in senescent cells, its expression is much more connected with diabetes, where it represents one of the commonly used markers for the detection of diabetes progression and its vascular complications. Therefore, we evaluated expression of PAI in tissues of HFD-fed mice. However, since the expression of PAI in aged mice was under the detection limit in all tested organs and since the results obtained with HFD-fed mice were non-significant, we removed these data from the updated version of the manuscript.

Figure 2- It concerns me that mitoTAM leads to a progressive decrease in body weight and it was not clear to me why this occur- is it due to decreased food consumption? It is interesting that other published senolytic agents did not result in decreased body weight in high fat diet- how do the authors explain this?

We thank the Reviewer for this important comment. After recalculating the food intake per mouse/day as suggested by Reviewer 3, significantly decreased food intake in MitoTam-treated mice as compared to HFD-fed controls became apparent. Therefore, part of the explanation of total weight loss is lower food consumption in MitoTam-treated HFD group. Another mechanism of

decreased body weight is due to the effect of MitoTam on the loss of adipose tissue mass, decreased level of triglycerides and decreased lipid deposition, which is partially caused by inhibition of adipogenesis (in adipose tissue) and by elimination of senescent cells known to accumulate lipids due to damaged mitochondria (in adipose tissue, liver and kidney) after MitoTam treatment. In relation to ‘other published senolytic agents’, we would like to stress that MitoTam acts differently and, unlike other senolytic agents, specifically targets mitochondria. This makes the agent highly innovative. We discuss this in more detail in the updated version of the manuscript to make it clearer for readers.

No error bars in SD controls (Figure 2a)

The error bars are present for all groups in Figure 2a. Both groups of SD mice had almost the same weight, therefore there are small error bars. We changed the graph to make the small error bars ‘visible’ in the updated version of the manuscript.

SFigure 2h – CD14 is not significantly increased and treatment didn’t reverse immune infiltration.

We thank the Reviewer for this comment. We agree that CD14 is not increased and we removed these data from the updated version of the manuscript.

Figure 2h – Lipid accumulation should be analysed and not only H&E images. Authors should include Nile Red, Oil Red O or Bodipy staining.

Concerning this point, H&E images represent a standard method to visualize lipid accumulation in organs, especially to detect NAFLD in liver. However, we performed additional staining with Oil Red O to support and quantify our results in the updated version of the manuscript. These data are now added as Supplementary Figure 2i, j.

SFigure 2k – Higher magnification images are needed and staining should be analysed.

We thank for this point. We ‘magnified’ the images and we completed the collagen analysis in the updated version of the manuscript, and these data are now shown as Supplementary Figure 2i, j.

Figure 3- Not clear in the text the rationale behind comparison of mitoTAM and Tamoxifen.

As we discussed above, we compared the effect of tamoxifen with MitoTam to see whether tamoxifen, as the biologically active component of MitoTam, is itself responsible for the observed effects of MitoTam or whether what we observe with MitoTam is dictated by its mitochondrial targeting. We discuss this in more detail in the updated version of the manuscript.

Figure 3h – Lipid accumulation should be analysed and not only H&E images. Authors should include Nile Red, Oil Red O or Bodipy staining.

Similarly as for Figure 2h, we carried out additional staining with Oil Red O to support and quantify our results, and this is now included in the updated version of the manuscript as Supplementary Figure 3g, h.

Figure 4f – Not sure what I am supposed to see here. This needs more explanation and analysis.

Figure 4f (now Figure 4e) shows microCT high 3D resolution image of *in vivo* scans, which provides comprehensive 3D visualization and quantification of total body fat mass before treatment (blue colour) and one month after treatment (red colour). Since each image represents overlap of two scans (before treatment and one month after treatment; scans were superimposed based on the position of mouse backbone and ribs), purple colour represents the overlap of adipose tissue present in the mouse during both scans. We understand this is not a commonly used method and we have explained it in more detail in the Figure legend in the updated version of the manuscript.

Reviewer #2 :

General:

The use of mitochondrially targeted tamoxifen (MitoTam, MT) for the treatment of obesity and type 2 diabetes via elimination of senescent cells is an interesting and quite novel approach. The data presented are thus of considerable interest although largely descriptive. However, a somewhat deeper scrutiny of possible mechanisms would be desirable.

We thank the Reviewer for his/her positive words about our manuscript, finding it of considerable interest. We agree that additional work, in particular concerning the mechanism, is needed, but we believe we are able to improve our experiments to make the manuscript acceptable for Nature Communication.

Given the fact that MT affects mitochondrial respiration by its interaction with respiratory complex 1, this immediately brings to mind the concept of mitohormesis which is used to explain the beneficial systemic effects of mitochondrial stress, and has attracted a lot of attention recently (for reviews see: Bar Ziv et al. 2020, doi: 10.15252/embr.202050094; Conte et al. 2020, doi: doi: 10.1007/s00281-020-00813-0; Klaus & Ost 2020, doi: 10.1016/j.exger.2019.110796). The systemic effects of MT are very similar to effects of the stress induced mitokines FGF21 and GDF15 (see also my specific comment below), so I think this is an important aspect which should be addressed.

This is a very important point, which can help us to clarify the mechanism of action of MitoTam and we thank the Reviewer for raising it. We addressed the role of mitohormesis and mitokines in the therapeutic effect of MitoTam treatment on the progress of T2DM in the updated version of the manuscript, as discussed in more detail at under the Reviewer's point 6.

Specific remarks:

1. The title might be a bit too strong as a statement. There was no real diabetes model investigated. C57BL/6N mice used here do become obese and insulin resistant when fed a high fat diet, but they never become really diabetic. Additional, maybe genetic diabetes models would need to be studied to draw that conclusion.

We respectfully disagree with the Reviewer's opinion that the C57BL/6 mouse is not appropriate model for human type 2 diabetes mellitus (T2DM) and that genetic models should be studied. Typically, patients with early stages of T2DM present with modest to moderate hyperglycemia (with glucose values similar to those in our model), hyperinsulinemia, obesity, fatty liver disease and insulin resistance. Furthermore, excessive calorie intake leading primarily to obesity and insulin resistance with gradual development of hyperglycemia represents a typical time-course of the development of human T2DM. The genetic models of diabetes such as *ob/ob* or *db/db* mice have much more severe hyperglycemia and insulin resistance as compared to what can be seen in typical patients with type 2 diabetes therefore representing more severe cases of poorly controlled diabetes

rather than general case of T2DM as seen in most patients. In humans, T2DM is a polygenic disease with very rare exceptions of patients with diabetes caused by a single gene mutation as it is the case for *ob/ob* or *db/db* mice. The fact that C57BL/6 is appropriate for studying T2DM has been documented in numerous papers (please see below).

Nevertheless, the Reviewer's point is well taken and we will explain in the Discussion that our results apply rather to early, less severe stages of T2DM and do not necessarily apply to all stages of diabetes.

<https://diabetes.diabetesjournals.org/content/37/9/1163.short>
<https://www.sciencedirect.com/science/article/pii/S0026049503005493#BIB20>
<https://www.jax.org/strain/000664>

2. Methods: The sex of the animals is not specified for all experiments. Considering that tamoxifen is a selective estrogen receptor modulator, it would be important to use both male and female mice (also considering possible side effects).

We thank the Reviewer for this comment. We used male mice because of their better hormonal stability and ability to induce obesity and T2DM when fed HFD. In many obesity/T2DM models it is very difficult and often not possible to induce obesity and T2DM in female mice while it can be easily achieved in male mice.

We agree that tamoxifen is a selective estrogen receptor (ER) modulator and that this must be taken into consideration in our experiments. On the other hand, the mechanisms of action of tamoxifen and MitoTam are different. In our previous experiments discussed in Rohlenova K et al. (Selective disruption of respiratory supercomplexes as a new strategy to suppress Her2^{high} breast cancer. *Antiox Redox Signal* 2017, 26, 84-103; reference #19 in our manuscript), we observed the same effect of MitoTam on elimination of cancer cells independently on their ER status (ER-wild type vs. ER-negative cells). Next, even tamoxifen (TX) is known to activate ER, we observed decreased activation of ER in cells treated with MitoTam (MT) (see picture attached below; MCF7 cells treated with 1 μ M MT or TX for 24 h). Moreover, we observed the same effect of MitoTam compared to tamoxifen on ER activation also *in vivo*, analysing the liver of HFD-fed mice treated with these agents twice per week for 4 weeks (see the picture attached below). Since the effect of tamoxifen on glucose level and loss of weight described in the literature is connected with activation of ER, all these results support different, ER-independent, mechanism of action of MitoTam.

Finally, there is the same effect of MitoTam on decrease of fasting glucose level detected in aged male and female mice (see the attached picture). This supports our statement that there is no side effect of MitoTam via ER. We included this reasoning/explanation in the updated version of the manuscript.

3. Results: The statement in lines 113-116 does not quite match the corresponding figure (1F,G). It is stated that aged mice showed an increased fasting glucose and poor glucose clearance. However, as presented in the figure these were not different from young CO. Aged MT treated mice rather showed very low glucose levels during the oGTT, which were even lower than in young-CO. To me this looks like an impairment of intestinal glucose uptake in the Aged+MT group rather than an improved glucose uptake. It would be helpful to also show insulin data during the oGTT. Also, data on food intake and body weight development during the MT treatment of aged mice would be of importance.

We agree with the Reviewer that data presented in Figure 1 are somewhat confusing and do not fully support our statement. The increase of glucose level in a group of aged mice is not as impressive as for our model of HFD-fed mice, however in this experiment we want to test whether there is any effect of MitoTam on glucose level to confirm the relevance of subsequent testing of MitoTam in the model of HFD-fed mice to study its ability to improve diabetic parameters.

Therefore, we have re-written our statement in the updated version of the manuscript to fit more with our experiments and ideas.

Since the data on food intake and body weight development during the MitoTam treatment of aged mice are of importance, we evaluated these parameters on a new cohort of aged mice. As shown in the pictures below we observed significant changes neither in body weight nor in food consumption.

4. Results: There is a discrepancy between the data on food intake presented in Suppl. Fig. 2B (no effect of MT on food intake on either HFD or LFD) and suppl Fig 4b (considerably reduced food intake by MT on HFD one month after treatment). Could this be due to long term side effects of MT?

As per the suggestion of Reviewer 3, we re-calculated food intake per mouse/day instead of 50 g of body weight/day. After this recalculation, a significant decrease in food intake in MitoTam treated HFD group by 27% as compared to HFD alone has become evident. It is also worth mentioning that a different method of food intake measurement was used in the prolonged effect experiment (Figure 4b). This measurement was performed in metabolic cages. In Supplementary Figure 2 we show food intake during MitoTam treatment of mice housed in standard cages. Taken together we do not think that prolonged effect of MitoTam on food intake can be considered a long-term “side effect” of MitoTam treatment as the mice were healthy throughout the course of the experiment. Part of the explanation of the prolonged effect could be due to the changes in adipogenesis that can last longer as compared to other mechanisms of action. One of the examples of such long-term effects that persist for months after withdrawal of the therapy is the treatment with PPAR- γ agonists such as rosiglitazone and pioglitazone.

5. Regarding the comparison between MT and TX treatment: in both treatments the same concentration (2mg/kg BW) was used. What was the rationale for using this concentration?

Since tamoxifen is the biologically active group of MitoTam, we used the same concentration of both agents to discern whether the observed effects are due to tamoxifen *per se* or whether they are dictated by mitochondrial targeting of MitoTam. We made this clearer in the updated version of the manuscript.

6. There is no suggestion or discussion about the specific mechanisms of MT compared to TX. It is known that mitochondrial stress leads to an activation of the integrated stress response via ATF4 resulting in increased expression of mitokines such as FGF21 and GDF15, which both have been shown to improve diabetes related features. It would be interesting to measure

their plasma levels and gene expression e.g. in liver, muscle or adipose tissue. In this respect it would also be of interest to look for possible browning of subcutaneous adipose tissue (e.g. UCP1 gene expression).

This is an important point and we thank the Reviewer for it. We have now focused more on the role of mitochondrial stress in the effects observed during MitoTam treatment in our model. We analysed the expression of FGF21 and GDF15 in tissues as well as the level of the FGF21 in plasma to better understand the mechanism of action of MitoTam, which is underlying its effect on reduction of obesity and alleviating the signs of T2DM. As shown below, we detected increased level of FGF21 in liver and subcutaneous adipose tissue (SAT) as well as in plasma of HFD-fed mice, which is in accordance with the literature, but we did not observe a significant decrease of this marker after MitoTam treatment. To conclude this experiment, FGF21 plays a role in diabetes induction/maintenance, but is not essential for the effect of MitoTam since there is no change in its expression despite improved diabetic parameters observed in treated animals.

Recent findings show a critical role of UCP1 in browning of white adipose tissue, resulting in increased glucose consumption and regulation of imbalance between energy intake and expenditure, which helps to fight obesity and metabolic dysregulation in patients with T2DM. This is a very interesting hint and we thank the Reviewer for it. As shown below and now also in Figure 2d, we detected significantly increased UCP1 expression in SAT in HFD-fed mice treated with MitoTam. This result correlates with our observation that adipocytes treated with MitoTam increase uncoupled respiration (see Figure 6k, j in new version of manuscript). All these experiments show a possible role of browning of adipose tissue after MitoTam treatment in improvement of diabetic parameters.

7. Regarding the strong effect of MT on body fat reduction in HFD fed mice (Fig. 2) I am wondering about possible effects on intestinal energy absorption (which could also explain the decreased glucose levels during oGTT after MT treatment in aged mice, Fig. 1). Since enterocytes have a very high metabolic activity, disturbance of their mitochondrial respiratory activity might have considerable effects on their activity relating to nutrient absorption. This could easily be addressed in a first step by measuring food assimilation, i.e. energy intake minus fecal energy excretion (measured by bomb calorimetry).

Since enterocytes have a very high metabolic activity, disturbance of their mitochondria and subsequently their activity would result in changes in total CO₂ production. Since there is no difference in respiratory exchange after MitoTam treatment (see Supplementary Figure 5a), the levels of CO₂ produced by mice can therefore indicate changes in intestinal energy metabolism. As shown below, we measured total CO₂ production by mice and we did not observe any changes after MitoTam treatment. We know, that this is not a direct prove of activity of the enterocytes, but it indicates little if any effect of MitoTam on these cells.

8. The considerable loss of body weight (Fig. 2A) of over 10 grams in 4 weeks is much higher than the loss of VAT (about 1 g). Even accounting for other fat depots this could also reflect a loss of lean body mass or possibly dehydration. Are there any data available on muscle mass or total lean mass? Was water intake affected by the MT treatment?

We thank the Reviewer for this very relevant question. As documented in Supplementary Figure 5d, we do not think that the loss of body weight is due to possible dehydration of mice since we did

not detect any significant changes in water consumption between control and MitoTam-treated animals. Similarly, as is now shown in Figure 4f MitoTam has no significant effect on lean mass. Our new results after recalculation of food consumption, as was suggested by Reviewer 3, rather show contribution of decreased food intake in MitoTam-treated mice as compared to HFD-fed controls, which partially explain total weight loss. Furthermore, increased UCP1 expression in SAT can also contribute to the total loss of body weight.

9. I do not understand Fig. 4D and the conclusion drawn. In the text it is stated (lines 212-213): “However, change of diet after the end of treatment (HFD to SD) stabilized the improved glucose uptake.” If the labeling corresponds to the labeling of the other subfigures, the HFD+MT group is actually missing, only the data for the SD+MT group are shown.

In Figure 4d we show glucose uptake in mice fed with HFD during the whole course of the experiment (first column), after which the HFD-fed mice were transferred onto standard diet after the end of the treatment for one month (the second column represents HFD-fed mice without any treatment transferred on SD, the third column represents HFD-fed mice treated with MitoTam and transferred on SD at the end of the treatment). We did this experiment to see whether change of diet helps to prolong the effect of MitoTam on glucose uptake, which was reverted back to the level found for mice one month after MitoTam treatment for mice fed with HFD even after the end of treatment (see Figure 4a). In this experiment, we did not include a group of MitoTam-treated HFD-fed mice staying on HFD for one month, since we acquired these data in Figure 4a. However, we are aware of some inconsistency of our results, especially the non-significant differences between non-treated and MitoTam-treated HFD-fed mice transferred onto SD. Therefore, we decided to remove these results from manuscript.

Also, I am wondering why for Fig. 4A an IP glucose tolerance test was performed and for Fig. 4D an oral GTT (similar to Fig 2)? Actually, it would be quite interesting to directly compare the results of an oGTT with that of an IP-GTT. If MT indeed affects intestinal nutrient (glucose?) uptake, the results could be quite different. Possible incretin effects would also be bypassed when using an IP-GTT.

Based on the complexity of experiments carried out to gauge the possible persistence of the MitoTam effect in HFD-fed mice even after the end of the treatment shown in Figure 4, these analysis were accomplished in collaboration with the Czech Centre for Phenogenomics, which routinely use the IP-GTT test. Even though they used a different method for glucose detection, we obtained the same data using the oGTT test (please, compare Figures 2d and 4a), which not only supports our glucose data by an independent, collaborating laboratory, but also indicates that MitoTam does not affect intestinal nutrient uptake.

10. Regarding the in vitro studies on 3T3-L1 adipocytes: How comparable are the in vitro and in vivo concentrations of MT and TX used?

The biodistribution of MitoTam (one application of 6 mg/kg for 24 h) assessed in different organs during our pre-clinical studies as a part of our clinical evaluation of MitoTam as an anti-cancer agent showed accumulation of 0.55 mg/kg of MitoTam in rat EAT. Similar results were observed also for pigs (as a part of a separate study). The concentration corresponds to the 0.67 μ M MitoTam used for *in vitro* treatment. Concentration of MitoTam administered at 2 mg/kg twice per week used in our *in vivo* experiments is therefore comparable to our *in vitro* experiments (corresponds to +/- 0.3 μ M). We included this information in the Methods section of the updated version of the manuscript to provide reasoning for the concentrations of MitoTam used in our study.

11. Discussion: The studies using 3T3-L1 adipocytes show nicely that MT directly affects adipocyte differentiation but seemingly has no or little effects on differentiated adipocytes. In my opinion, this does not quite support the conclusion that MitoTam reduces visceral adipose tissue by suppressing adipogenesis since no adipogenesis is necessary for initial adipocyte hypertrophy. Furthermore, adipogenesis does not really play a role in the weight reduction effects in already obese mice. Most importantly, TX basically showed the same fat reducing effect as MT in vivo without affecting adipogenesis in vitro, suggesting other mechanism must be involved.

We agree with the Reviewer that suppressing adipogenesis does not fully explain the effect of MitoTam on adipose tissue weight reduction. After recalculating the food intake per gram of body weight as suggested by Reviewer 3, significantly decreased food intake in MitoTam-treated mice as compared to HFD-fed controls became apparent. This indicates that reduced adipocyte hypertrophy due to lower food intake is another important player in overall changes in body weight.

As far as a possible role of adipogenesis in weight loss is concerned, the development of obesity is characterized by the combination of the increase in adipocyte size (hypertrophy) and in their number (hyperplasia). At present, adipogenesis during obesity has also been shown to affect the composition and regeneration of adipose tissue (Trends Endocrinol Metab. 2016, 27, 574-585; doi: [10.1016/j.tem.2016.05.001](https://doi.org/10.1016/j.tem.2016.05.001)). 3T3-L1 cells present a suitable and best studied model of adipogenesis. For this reason, we decided to use this cell line to determine the effect of MitoTam on adipogenesis. One of the evaluations of a new molecule with influence on adipose tissue and obesity is to determine its effect on adipogenesis. During *in vitro* adipogenesis in the adipocyte maturation phase, adipocyte proliferation is stopped and lipid accumulation occurs (which is similar to adipocyte hypertrophy). In this maturation phase, MitoTam had also a strong effect on accumulation of lipids, but the same results were not observed during tamoxifen treatment. Moreover, we newly show the effect of MitoTam also on elimination of mature, senescent adipocytes (see new Figure 5), which can contribute to the loss of adipose tissue weight. Therefore, we think that the reduction of body weight is partially caused by combination of inhibition of adipogenesis and elimination of hypertrophic adipocytes.

The role of tamoxifen in improvement of glycemia and fat reduction was described in the literature. Related to this, the effect of tamoxifen is strongly connected with activation of the estrogen receptor (ER). As we show above (see Figures at Point 2), our modification of tamoxifen with TPP changed its mode of action to be ER-independent. Thus, even though we observed similar effect of both agents on loss of adipose tissue mass, this can be caused by different mechanisms – MitoTam-induced decrease of adipogenesis and elimination of senescent hypertrophic adipocytes versus tamoxifen-induced activation of ER. Importantly, unlike tamoxifen, which only decreased hyperglycemia and reduced adipose tissue mass, MitoTam shows a more complex effect on the improvement of diabetic parameters, including elimination of senescent cells, improvement of T2DM comorbidities or suppression of systemic inflammation. Furthermore, we observed prolonged effect of MitoTam on weight reduction in comparison with tamoxifen, where only transient effect followed by over-compensation resulting in increased fat mass was described (Sheean et al, 2012, doi.org/10.1007/s10549-012-2200-8; Ali et al, 1998, [doi.org/10.1016/S0969-8043\(97\)00082-1](https://doi.org/10.1016/S0969-8043(97)00082-1); Hesselbarth et al, 2015, doi.org/10.1016/j.bbrc.2015.07.015).

12. The fact that TX affected body mass and fat in the same way as MT but did not decrease insulin levels or B-gal positive cells (Fig. 3), suggests that the improvement of glucose metabolism is mainly due to the senolytic effect of MT rather than its effect on adipose tissue mass. This is a highly interesting aspect which should be addressed in more detail. It also makes me wonder why a major focus was put on adipose tissue and adipogenesis. Wouldn't it be more appropriate to focus more on liver or beta-cell function? Also, muscle might be

important considering its prominent role in glucose uptake and contribution to overall energy metabolism.

Most of the factors affecting progression of T2DM are tightly connected with adipose tissue (e.g. increased expression of pro-inflammatory factors or hormones, which can induce cellular damage or senescence in distal organs such as liver or pancreas). In fact, we decided to study the effect of MitoTam on T2DM since this pathology is characterised by high level of senescent cells in particular in adipose tissue (as highlighted, e.g., in the “Munoz-Espin and D Serrano M. Cellular senescence: from physiology to pathology. Nat Rev Mol Cell Biol 2014, 15, 482-496” paper that is included as reference #10 in our manuscript). Changes in this tissue, which include not only loss of adipose mass but mainly functional changes, therefore represent an important mechanism in the regulation of diabetic parameters.

Reviewer #3:

In this manuscript, Vacurova et al explore the impact of mitochondrially targeted tamoxifen (MitoTam) in a variety of metabolically compromised mouse models. They report that MitoTam improves a number of metabolic parameters and conclude that this is primarily due to effects to reduce senescent cell numbers and to impede adipogenesis. While there is some interesting data in this manuscript, there are a number of significant issues that require attention.

There are a substantial number of instances throughout the manuscript where non significant changes are referred to in a manner that would suggest they are different (both mouse data and human data). The data are either significant or they are not. It is inappropriate to make conclusions based on trends (often weak) and the authors should remove such statements from the manuscript.

We agree with the Reviewer and we have performed additional experiments to reach significance for results where we saw a trend but we did not observe significant differences. Additional experiments are included in the updated version of the manuscript, non-significant results have been removed.

The assertion that senescent cells are responsible for many of the metabolic defects in the various mouse models and hence the reduction of senescent cells by MitoTam is a major mechanism underpinning its effects is not especially well supported by the data. For example, in Figure 1 there are substantial differences in senescence markers between the young and old control groups, but the glucose tolerance test between these groups is not statistically different. On the other hand the aged group treated with MitoTam has a very pronounced improvement in glucose tolerance, but shows predominantly non significant changes in senescence markers, particularly in important metabolic tissues such as the liver.

We thank the Reviewer for this comment. We have performed additional experiments with aged mice, which have been added to Figure 1 in the updated version of the manuscript. Increased number of mice per group in experiments shown in Figure 1 have now allowed to reach significance and have helped to reduce inconsistency between individual panels.

As mentioned above, aging of organism increases the probability of development of age-related diseases, like diabetes and its related comorbidities. The results observed in our aged mice correspond more to the pre-diabetic state, where hyperglycemia is not fully developed to the level observed for T2DM more relevant model of HFD-fed mice. However, the aim of this experiment was to test whether there is an effect of MitoTam on glucose levels in aged animals to confirm relevance of subsequent testing of MitoTam on more relevant model of HFD-fed mice to study its

ability to improve diabetic parameters. Therefore, we have re-written our statement in the updated version of the manuscript to fit more with our experiments and ideas.

Likewise in Figure 3, the tamoxifen group exhibits substantially reduced adiposity and improved glucose tolerance compared to the control group and yet the senescence markers are not different between these groups. These data do not strongly support a causative role for senescent cells in the induction of metabolic defects and thus call into question one of the main mechanisms proposed to underpin the effects of MitoTam. The authors need to address these inconsistencies.

As we discuss in our manuscript, we do not think that elimination of senescent cells is the only reason for anti-diabetic effects of MitoTam. Rather, we propose that our results are given by the combination of the effect of MitoTam, with the biologically active tamoxifen moiety targeted to mitochondria affecting the glucose parameters and eliminating senescent cells (or reducing their number), also involving the inhibitory effect of MitoTam on adipogenesis and perhaps other mechanisms leading to decreased body weight. We also observed considerable improvement of diabetes- and obesity-related comorbidities, such as attenuation of fat deposition in the liver and in the kidneys. We discuss the mechanism including the effect of tamoxifen alone on glucose parameters as well as on adipose tissue in more detail in the updated version of the manuscript in order to make this clear for the reader.

The authors report that there is no change in energy expenditure and no change in food intake – so how is it that the treated mice are losing weight (and fat mass)? The issue is that food intake in this manuscript has been reported normalised to body weight, which largely obscures the fact that the animals treated with MitoTam must be eating less total calories (as they are losing body weight). Energy expenditure has not been normalised to body weight and thus there is an imbalance in these measures. The authors are referred to an excellent review article on assessment of mouse energy metabolism (Nat Methods. 2011, 9: 57–63) where the recommendation is that the analysis of energy balance should be made on the raw data (e.g. calorie intake) expressed per mouse (rather than normalising one side of the equation by body weight and not the other). It is likely that much of the benefit of MitoTam is in fact a result of an acute drop in caloric intake, which is known to rapidly improve metabolic parameters in mice. A pair feeding study would help to resolve this.

We thank the Reviewer for this point. We have re-calculated food consumption data to gram of food/mouse/day as suggested. Using two way ANOVA, there was a significant decrease of food intake in MitoTam-treated HFD mice (by 27 %). As we discuss further in the manuscript, the decrease in the body weight is also a combination of the effect of MitoTam leading to the loss of adipose tissue mass, decreased level of triglycerides and decreased lipid deposition, which is partially caused by inhibition of adipogenesis (in adipose tissue) and by elimination of senescent cells known to accumulate lipids due to damaged mitochondria (in liver, adipose tissue and kidney). Additionally, we calculated also the contribution of the loss of lean body mass from our microCT data to acquire complex analysis of changes in body weight, but we did not observe significant changes (see Figure 4f). Instead of this, subsequent decreased food consumption (Supplementary Figure 2b) as well as increased UCP1 expression in SAT (Figure 2d) can contribute to the total loss of weight as we discuss in answers to Reviewer 1 and 2. All these new findings are discussed in the updated version of the manuscript.

Following on from the above, the authors provide evidence in in vitro experiments that MitoTam influences adipogenesis and propose this might be responsible for the weight (and fat) loss phenotype observed. The concern with this proposal is that the authors have used a model of ‘reversing’ diet induced defects – namely mice have been fed a high fat diet long

term and are then exposed to MitoTam for a short 4 week period. The authors own data (Figure 2A) shows that body weight has stabilised with the long term fat feeding regime and thus under these conditions there is unlikely to be extensive adipogenesis occurring. Hence the proposal that MitoTam is blocking adipogenesis to cause the observed weight loss over the 4 week treatment period is difficult to reconcile. Please comment on this discrepancy and how it impacts on the conclusions put forward in the discussion.

We agree with the Reviewer that suppressing adipogenesis does not fully explain the effect of MitoTam on adipose tissue weight reduction. After re-calculating the food intake per gram of body weight, significantly decreased food intake in MitoTam-treated mice as compared to HFD-fed controls became apparent. This indicates that reduced adipocyte hypertrophy due to lower food intake is another important player in overall changes in body weight.

As far as a possible role of adipogenesis in weight loss is concerned, the development of obesity is characterized by the combination of the increase in adipocyte size (hypertrophy) and in their number (hyperplasia). Recently, adipogenesis during obesity has also been shown to affect the composition and regeneration of adipose tissue (Trends Endocrinol Metab. 2016, 27, 574-585; doi: [10.1016/j.tem.2016.05.001](https://doi.org/10.1016/j.tem.2016.05.001)). One of the evaluations of a new molecule with an effect on adipose tissue and obesity is to determine its effect on adipogenesis. 3T3-L1 cells present a suitable and best studied model of adipogenesis. For this reason, we decided to use this cell line to determine the effect of MitoTam on adipogenesis. We have now focused on the effect of MitoTam on mature adipocytes that show markers of senescence. As documented in new Figure 5, MitoTam not only prevents adipogenesis, but also eliminates mature, senescent adipocytes, which can contribute to the loss of adipose tissue weight.

It is hard to gauge from the manuscript which tissues might be responsible for the benefits of MitoTam on glucose tolerance and insulin sensitivity, as many tissues have been assessed for different parameters (i.e. lipid accumulation, senescence markers). The authors should consider conducting hyperinsulinemic/euglycemic clamps to determine if specific tissues (e.g. liver, adipose) may have a more major role in driving the beneficial effects on insulin sensitivity.

We do not think that there is one predominant tissue that affects all diabetic parameters. We propose that it is more likely a combined effect of MitoTam on different organs and tissues leading to improvement of glucose tolerance and insulin sensitivity. An important role in this mechanism is played by adipose tissue producing a number of factors affecting progression of T2DM (e.g. increased expression of pro-inflammatory factors or hormones, which can induce cellular damage or senescence in distal organs such as liver or pancreas). Changes in these tissues, which include not only the loss of adipose tissue mass but also functional changes, therefore represent an important mechanism in regulation of glucose parameters. Moreover, elimination of damaged and senescent cells from liver as well as decreased accumulation of lipids in this tissue after MitoTam treatment also contribute to improved glucose uptake. We discuss it in more detail in the updated version of the manuscript.

Minor points

The statistical analysis section indicates one way ANOVA was used for all mouse studies. For the data in Figure 2, a 2 way ANOVA is likely more appropriate, with diet and treatment as the 2 main factors.

We agree and have changed the statistical analysis used in this panel in the updated version of the manuscript.

The authors may consider calculating the incremental area under the curve for the glucose tolerance tests rather than the total area under the curve – it might reveal more substantial differences for some of the experiments.

With respect to Reviewer's point, calculation of total area under the curve represents a standard method used for statistical evaluation.

The authors frequently present histological images to highlight lipid accumulation and then complement this with the more robust biochemical assessment of triglyceride levels. They are encouraged to provide a triglyceride measurement for the samples depicted in Figure 4I, to provide continuity for the manuscript and greater confidence in the reported finding.

We evaluated TAG level in liver tissue, and the new results are now included in the updated version of the manuscript as Supplementary Figure 4f.

Line 62 in the introduction implies that ectopic lipid accumulation in non-adipose tissues only occurs once fat storage is exhausted. This is not accurate, as lipid accumulation and insulin resistance occur in peripheral tissues very rapidly on exposure to excess dietary fat (i.e. within days). Please correct this statement.

This is a very good point and we thank the Reviewer for it. We corrected this statement in the Introduction part of the updated version of the manuscript.

Line 130 – it is suggested that the fat-fed C57BL/6 mouse model is a model of diabetes, but in fact this mouse strain will become obese and insulin resistance, but never fully develops diabetes. Please correct.

We respectfully disagree with Reviewer's opinion that C57BL/6 is not appropriate model for human type 2 diabetes mellitus (T2DM). Typically, patients with early stages of T2DM present with modest to moderate hyperglycemia (with glucose values very similar to those observed in our model), hyperinsulinemia, obesity, fatty liver disease and insulin resistance. Furthermore, excessive calorie intake leading primarily to obesity and insulin resistance with gradual development of hyperglycemia represents a typical time-course of the development of human T2DM. The fact that C57BL/6 is appropriate for studying T2DM has been documented in numerous papers (please, see below).

Nevertheless, the Reviewer's point is well taken and we are happy to explain in the Discussion that our results apply rather to early less severe stages of diabetes and do not necessarily apply to all stages of the pathology.

<https://diabetes.diabetesjournals.org/content/37/9/1163.short>
<https://www.sciencedirect.com/science/article/pii/S0026049503005493#BIB20>
<https://www.jax.org/strain/000664>

The relevance of Figure 4D is questionable, as the impact of changing from HFD to SD appears to be the overriding factor in these effects, not previous treatment with MitoTam.

We agree with the Reviewer and have removed Figure 4d from the updated version of the manuscript.

Line 335 – the authors indicate that tamoxifen alone had no effect on lipid accumulation, but this is incorrect as per the data shown in Figure 3B and extended Figures 3B and 3G.

We agree with the Reviewer that there is a partial effect of tamoxifen on lipid accumulation and we have re-written this statement in the updated version of the manuscript.

REVIEWER COMMENTS

Reviewer #1 (Remarks to the Author):

The authors have addressed most of my concerns. However, I don't agree with their explanation that markers of senescence are expressed inconsistently. It is true that senescent cells should be detected by a combination of different markers as there is no universal marker. But this still means that the same assessments should be performed in all the tissues and all the results should be shown and not only individual selected results depending on significance.

It would help clarity to call beta-Gal staining SA-beta-Gal staining throughout the manuscript as done in material and methods.

As the authors have now stated that mice with MitoTAM treatment consumed significantly less than Mito treated mice I wonder how much of the effect is due to treatment versus reduced calorie intake. As shown before calorie restriction decreases lipid accumulation, decreased lipid content alters and lowers the SASP and calorie restriction leads to a decrease in senescent cells. How can the authors distinguish if the changes shown are due to the treatment or calorie reduction?

Reviewer #2 (Remarks to the Author):

The manuscript has gained considerably by the revision. Major concerns have been addressed and additional data presented. However, there are still a number of points I am still not sure about.

Specific remarks:

1. My remark concerning glucose homeostasis in aged mice as shown in Fig.1 was not really addressed. It is still stated lines 117-121: "Aged mice also showed slightly increased fasting glucose, poor glucose clearance assessed by oral glucose tolerance test (oGTT) (Fig. 1d, e) Treatment with MitoTam improved all these parameters in aged mice and shifted them to levels seen in young animals (Fig. 1d-f), indicating plausibility to study MitoTam in a relevant diabetic model." This is not evident from the data shown. Therefore, the title of this figure does not reflect the data and should be corrected. Blood glucose was not significantly elevated in aged mice (Fig. 1d) and glucose clearance after the oGTT was the same in young and aged mice (Fig. 1e). As I remarked before, aged MT treated mice rather showed very low glucose levels during the oGTT, which were even lower than in young-CO. This still looks like an impairment of intestinal glucose uptake in the Aged+MT group rather than an improved peripheral glucose uptake (see also 6.). It is stated in the rebuttal that a new cohort of aged mice was investigated. Fig. 1F however, is still the same.

2. Regarding the re-evaluation of the food intake data, it becomes evident that reduction of food intake by TM is a major mechanism leading to decreased adiposity. A reduction of 27% is quite considerable. These data should therefore be presented in the main fig. 2 and not "hidden" in the supplements. It makes little sense to point out that food intake is lower on the HFD than on the LFD because of the higher energy content of the HFD (lines 143-148) since this is obvious. It would be more interesting to state and compare the energy intake rather than food intake. How much of body fat change could,

theoretically, be explained by the 27% reduced food intake in the MT group? This could easily be calculated and would show if the reduction in food intake was sufficient to explain the differences in fat mass.

3. What is the difference between Suppl Fig. 4b and suppl. Fig 5d? Both show food intake of mice treated i.p. twice per after or one month after treatment. But there are different data shown. Are these from different cohorts? In suppl fig 4b there is a reduction in cumulative food intake one month after treatment, in Fig. 5d not. Why show cumulative food intake? It would be more informative to show the daily pattern of food intake, analogous to the data on activity and RER. Minor point: in suppl fig 4b the labeling of the x-axis is wrong (this cannot indicate hours).

4. Regarding the effect of TM on food intake: A reduction in food intake can have very different reasons, an obvious one being the induction of nausea or visceral malaise. Are there any indications from your human trials about nausea, sickness and/or abdominal pain as a side effect? This should at least be discussed.

5. Regarding the strong effect of MT on reduction of body fat and circulating glucose I am still wondering about possible effects on intestinal energy absorption. The authors argue that this cannot be the case because overall CO₂ production would be affected. I do not really follow this argument. I would strongly urge to check diet digestibility or measure feces energy/glucose content. At least the possibility should be discussed. Also, when comparing the results of the oGTT (suppl fig. 2c) and IP-GTT (suppl. Fig 4a) it is obvious that they differ considerably. In the HFD-MT group glucose levels are always lower than in the HFD-CO after the oGTT but basically show the same pattern, whereas after the IP-GTT they reach the same levels after 15 minutes and then decline more rapidly. For me this is a substantial difference which could be an indication that intestinal glucose absorption or metabolism might be affected by TM.

6. I appreciate that the authors investigated a possible induction of FGF21 and GDF15 by MitoTam. Since this is apparently not the case, maybe a sentence relating to this (referring either to unpublished data or showing the data as a supplement) could be included in the manuscript.

Reviewer #3 (Remarks to the Author):

The manuscript revisions conducted by the authors have provided some improvements, however there are still a number of significant issues that require addressing.

In Figure 1, there are very large differences between young and old mice in senescence markers, but no significant difference in the measures of glucose metabolism. This does not support a major role for senescent cells in inducing age-related metabolic dysfunction and thus, although there are effects of MitoTam on both parameters, it is difficult to assign any causality. It is also worth noting that the senescence measures have not been conducted in adipose tissue, which is the focus of this manuscript.

In addition, in relation to Figure 1 the fasting glucose levels and the glucose tolerance changes between

young and old mice are still described in a manner that implies there is a significant difference (lines 117-118), when in fact there is not - please correct.

The authors have stridently rebutted the comments of 2 x reviewers that their mouse models is not a true model of T2DM. The mild increase in fasting glycemia in fat-fed animals is a common observation in mice that are insulin resistant, and is somewhat akin to a state of pre-diabetes. However it is not T2DM, and the authors have no evidence for beta cell dysfunction in their animals (an essential component of T2DM), so the mice should be referred to simply as obese and insulin resistant and not diabetic.

As noted in the previous review, once the data relating to Figure 2 was presented appropriately, it is clear that there is a marked (27%) drop in calorie intake in the HFD mice receiving MitoTam. Calorie restriction is known to induce a raft of favorable metabolic effects, such as improve glucose tolerance, reduce lipid accumulation in non-adipose tissues and lower systemic inflammation. MitoTam treatment induced these same metabolic effects in mice, as well as eliminating senescent cells (in vivo and in vitro) and impairing adipogenesis (in vitro). What is impossible to reconcile is the relative importance of the direct effects of MitoTam (on senescent cells and adipogenesis) vs. the reduction in calorie intake induced by this compound. A pair feeding study would resolve this and should be undertaken.

Further to this point, the abstract completely fails to acknowledge the large decrease in food intake induced by MitoTam, and the fact that this likely mediates some of the favorable metabolic effects of MitoTam. Please correct this.

I fail to see the relevance of the change in UCP1 mRNA in subcutaneous adipose tissue. Firstly, while UCP1 protein is clearly important in thermogenesis, there are a raft of other changes (morphological and molecular) that occur during browning of adipose tissue, and without measurement of some of these parameters it is inappropriate to be so definitive about the browning of adipose tissue in response to MitoTam. Notwithstanding this issue, the other major concerns are 1) the browning of adipose tissue is suggested to favourably impact metabolic homeostasis primarily by increasing energy expenditure. The authors assert (lines 386-387) that there is a role for adipose browning in mediating the effects of MitoTam on weight loss, yet their own indirect calorimetry data shows no change in energy expenditure in response to MitoTam treatment. Please clarify how these findings can be reconciled? 2) calorie restriction is known to induce adipose browning (e.g. Fabbiano et al *Cell Metab*, 24: 434-446, 2016), and given that there was no change at all in UCP1 expression in the SD mice that received MitoTam, it is highly plausible that any potential change in adipose browning is being induced by the reduction in calorie intake in the HFD group and not directly by the MitoTam.

There are still instances where non-significant changes are being described in a manner that implies significance (e.g. line 192-194). This must be addressed.

First of all, we thank the Reviewers for insightful comments and queries. We understand that responding to the points they have raised is of importance in order to bring the manuscript to high standard so that it is publishable in Nature Communication. We would like to point out that due to the complexity of the experiments, we have changed the colour coding of individual experimental groups, which should make the results easier to be navigated through. Of note, during the revision period, one of the co-authors (Sona Hubackova) was married and is now listed under the name of Sona Stemberkova Hubackova.

Reviewer #1 (Remarks to the Author):

The authors have addressed most of my concerns. However, I don't agree with their explanation that markers of senescence are expressed inconsistently. It is true that senescent cells should be detected by a combination of different markers as there is no universal marker. But this still means that the same assessments should be performed in all the tissues and all the results should be shown and not only individual selected results depending on significance.

Based on the recommendation of the Reviewer we included all evaluated senescent markers in the revised version of the manuscript. To be consistent, we now show the combination of SA- β -gal with p21 for aged mice and SA- β -gal with p16 for HFD mice as the most expressed genes in respective models in the primary Figures (1, 2, 3 and 4). Other senescent markers measured for both models in respective organs are now shown in Supplementary Figures 1, 2 and 3.

It would help clarity to call beta-Gal staining SA-beta-Gal staining throughout the manuscript as done in material and methods.

We thank the Reviewer for this point. We altered this accordingly in the revised version of the manuscript.

As the authors have now stated that mice with MitoTAM treatment consumed significantly less than Mito treated mice I wonder how much of the effect is due to treatment versus reduced calorie intake. As shown before calorie restriction decreases lipid accumulation, decreased lipid content alters and lowers the SASP and calorie restriction leads to a decrease in senescent cells. How can the authors distinguish if the changes shown are due to the treatment or calorie reduction?

We agree with the Reviewer that reduced calorie intake can influence all mentioned parameters, thus it may be difficult to discern the direct effect of MitoTam. To allow for clear interpretation of the results, we performed a pair-feeding experiment. For this purpose, we divided a cohort of HFD-fed mice into 3 groups. The first group was the control and received food ad libitum, the second group of mice was treated with MitoTam and was given food also ad libitum, the third group was treated only with the excipient, but the amount of food provided to this group matched food consumed by the second group (see Materials and Methods). As now shown in Figures 2, 3 and Suppl. Figure 2 in the revised version of the manuscript, with the same calorie intake MitoTam-treated mice decrease body weight, amount of VAT, blood glucose and senescent markers more than mice with reduced food only. This experiment therefore shows a direct effect of MitoTam on the assessed parameters.

Reviewer #2 (Remarks to the Author):

The manuscript has gained considerably by the revision. Major concerns have been addressed and additional data presented. However, there are still a number of points I am still not sure about.

Specific remarks:

1. My remark concerning glucose homeostasis in aged mice as shown in Fig.1 was not really addressed. It is still stated lines 117-121: "Aged mice also showed slightly increased fasting glucose, poor glucose clearance assessed by oral glucose tolerance test (oGTT) (Fig. 1d, e) Treatment with MitoTam improved all these parameters in aged mice and shifted them to levels seen in young animals (Fig. 1d-f), indicating plausibility to study MitoTam in a relevant diabetic model." This is not evident from the data shown. Therefore, the title of this figure does not reflect the data and should be corrected. Blood glucose was not significantly elevated in aged mice (Fig. 1d) and glucose clearance after the oGTT was the same in young and aged mice (Fig. 1e). As I remarked before, aged MT treated mice rather showed very low glucose levels during the oGTT, which were even lower than in young-CO. This still looks like an impairment of intestinal glucose uptake in the Aged+MT group rather than an improved peripheral glucose uptake (see also 6.). It is stated in the rebuttal that a new cohort of aged mice was investigated. Fig. 1F however, is still the same.

We thank the Reviewer for this point. First, we would like to mention that additional analysis performed in the course of the rebuttal were accomplished using frozen organs from aged mice acquired over longer period of time, since we did not have 18 month old mice in our animal house at the same period. Therefore, some analyses were missing. During the revision we performed additional experiments with a new cohort of aged mice, which are now added to Figure 1 in the revised version of the manuscript. Increased number of mice per group in experiment shown in Figure 1f now allowed us to achieve significance for fasting glucose level, which now corresponds to our statement addressed in lines 117-121. We agree that oGTT performed in young and aged mice is not much different. This small difference between young and aged animals was described in the literature and reflects, for example, lower adipose tissue accumulation in aged animals compared to HFD-fed mouse model, which is one of the major tissues regulating the level of blood glucose (doi.org/10.2337/db19-0377). Anyway, clearance of glucose from blood is faster in young animals, indicating impaired metabolism in aged animals. However, we agree that in the context of our article these data do not support our message. We therefore decided to remove oGTT from Fig. 1 of the revised version of the manuscript and to correct the title.

We are aware of the lack of leptin analysis, which was caused by unavailability of a new cohort of aged mice as well as of the Luminex and Elisa kits. Additional leptin analysis is now shown in Fig. 1g of the revised version of the manuscript. Similarly, the insulin Luminex kit as well as Elisa kit are still unavailable and we are unfortunately not able to analyze an additional cohort of mice. You can see the original analysis in the graph attached below, showing increased insulin level in plasma of aged mice, which has tendency to decrease after MitoTam treatment. Because of the lack of a significant difference, we did not included these data in the revised version of our manuscript. Notwithstanding this, all these data support our statement that MitoTam improves the impaired metabolism in aged mice.

Even though senescence and aging were described to contribute to the development of age-related diseases like diabetes, this model is not equal to fully developed diabetes. The goal of this pilot experiment was to find whether MitoTam has an effect on metabolic parameters, which will be then

studied using an appropriate model of obesity and obesity-induced prediabetic state. We discuss this in the revised version of manuscript to make this clear for readers.

We also agree, that effect of MitoTam on glucose clearance is bigger compared to young animals. Based on our new results shown in Fig. S1f we do not think, that this is an effect of impairment in intestinal glucose uptake, since we did not observed changes in food intake as well as faeces glucose content (see picture below). However, as we discuss below for the model of HFD fed mice the mechanism of glucose clearance represents a complex and important mechanism of action of MitoTam, which will be study separately in further project.

2. Regarding the re-evaluation of the food intake data, it becomes evident that reduction of food intake by TM is a major mechanism leading to decreased adiposity. A reduction of 27% is quite considerable. These data should therefore be presented in the main fig. 2 and not “hidden” in the supplements.

This is a relevant comment and we thank for it. In the revised version of the manuscript, you can find these data in Fig 2C. We also re-evaluated the data to be expressed in kcal/day, which corresponds to the standard way used in the literature.

It makes little sense to point out that food intake is lower on the HFD than on the LFD because of the higher energy content of the HFD (lines 143-148) since this is obvious. It would be more interesting to state and compare the energy intake rather than food intake.

As suggested, all food intake is now expressed in kcal/day in the revised version of the manuscript.

How much of body fat change could, theoretically, be explained by the 27% reduced food intake in the MT group? This could easily be calculated and would show if the reduction in food intake was sufficient to explain the differences in fat mass.

Theoretically the reduction in food intake can fully explain changes in body fat and body weight. However, because we do not have information about total fat mass and total fat free mass before and after the treatment, such calculation would be only an estimate. To better address the question how much of the differences can be attributed to food intake reduction and how much is caused by other mechanisms, we performed a pair-feeding experiment. Results of this experiment overall show that

both, food intake reduction and other mechanisms contribute to MitoTam action. Please see Figures 2, 3 and Supplementary Figure 2 of the revised version of the manuscript.

3. What is the difference between Suppl Fig. 4b and suppl. Fig 5d? Both show food intake of mice treated i.p. twice per after or one month after treatment. But there are different data shown. Are these from different cohorts? In suppl fig 4b there is a reduction in cumulative food intake one month after treatment, in Fig. 5d not. Why show cumulative food intake? It would be more informative to show the daily pattern of food intake, analogous to the data on activity and RER. Minor point: in suppl fig 4b the labeling of the x-axis is wrong (this cannot indicate hours).

Supplementary Figure 5d originally showed water intake, food intake was shown only in Supplementary Fig 4b. In the revised version of the manuscript, the data from the indirect calorimetry study are presented in an updated form. Daily patterns of all parameters are shown next to the 24 h mean values with statistics; total energy expenditure is not expressed per kg of body weight unlike in the previous version of the manuscript, instead regression between total energy expenditure and body weight is presented in the revised form of the manuscript, in order to; to avoid the confusion when all data from this measurement are in one figure panel, and water intake is not shown (as it is not relevant). Please see Supplementary Figure 5 in the revised form of the manuscript.

4. Regarding the effect of TM on food intake: A reduction in food intake can have very different reasons, an obvious one being the induction of nausea or visceral malaise. Are there any indications from your human trials about nausea, sickness and/or abdominal pain as a side effect? This should at least be discussed.

In our clinical study, we observed nausea only in three patients out of the cohort of 65 included in phase 1 (short administration) and phase 1b (repeated administration). We have not observed any other serious deleterious effects, abdominal pain or loss of appetite. Since our clinical trial included patients in terminal stage of cancer, these three cases of nausea can reflect acute condition of the patients. We now discuss this observation in the revised version of the manuscript.

5. Regarding the strong effect of MT on reduction of body fat and circulating glucose I am still wondering about possible effects on intestinal energy absorption. The authors argue that this cannot be the case because overall CO₂ production would be affected. I do not really follow this argument. I would strongly urge to check diet digestibility or measure feces energy/glucose content. At least the possibility should be discussed. Also, when comparing the results of the oGTT (suppl fig. 2c) and IP-GTT (suppl. Fig 4a) it is obvious that they differ considerably. In the HFD-MT group glucose levels are always lower than in the HFD-CO after the oGTT but basically show the same pattern, whereas after the IP-GTT they reach the same levels after 15 minutes and then decline more rapidly. For me this is a substantial difference which could be an indication that intestinal glucose absorption or metabolism might be affected by TM.

We agree that this point deserves more focus in order to explain the role of MitoTam in glucose lowering. As proposed by the Reviewer, we evaluated faeces glucose content in HFD-fed mice treated with MitoTam during the pair-feeding experiment. MitoTam-treated mice (MT) tend to increase glucose content in faeces in comparison with control mice fed with the comparable amount of food (PF; mice fed ad libitum were used as a control). However, we did not observe a significant difference. These data are now in Supplementary Figure 2i and discussed in the revised version of the manuscript. Furthermore, we did not observed any morphological changes in the duodenum and small intestine during our preclinical experiments using rats. Since glucose is not a part of the diet, its increased level in

faeces after MitoTam treatment can reflect not only decreased intestinal absorption, but also increased cleavage of poly- or oligosaccharides. Compared to significant decrease of glucose level in blood, its decreased absorption therefore cannot fully explain the effect of MitoTam on glucose lowering.

However, we agree that detailed clarification of the effect of MitoTam on metabolism and intestinal absorption present an important part of the mechanism of action of MitoTam and will be the subject of our further research.

6. I appreciate that the authors investigated a possible induction of FGF21 and GDF15 by MitoTam. Since this is apparently not the case, maybe a sentence relating to this (referring either to unpublished data or showing the data as a supplement) could be included in the manuscript.

We thank the Reviewer for this point. We discuss these data in the revised version of the manuscript.

Reviewer #3 (Remarks to the Author):

The manuscript revisions conducted by the authors have provided some improvements, however there are still a number of significant issues that require addressing.

In Figure 1, there are very large differences between young and old mice in senescence markers, but no significant difference in the measures of glucose metabolism. This does not support a major role for senescent cells in inducing age-related metabolic dysfunction and thus, although there are effects of MitoTam on both parameters, it is difficult to assign any causality. It is also worth noting that the senescence measures have not been conducted in adipose tissue, which is the focus of this manuscript.

With all due respect to the Reviewer, presence of senescent cells in organism is a well known 'trigger' or mediator of many age-related diseases including diabetes, which was described in a number of compelling publications (doi: [10.2337/db14-1820](https://doi.org/10.2337/db14-1820); doi: [10.1016/j.tcb.2020.07.002](https://doi.org/10.1016/j.tcb.2020.07.002); doi: [10.1038/nrd.2017.116](https://doi.org/10.1038/nrd.2017.116)). One mechanism described how components of the SASP secreted by senescent cells contributes to the increase of glucose level in blood and confers insulin resistance, which alter the overall metabolism and contributes to the development of diabetes. We performed additional experiments with aged mice, which are now added to Figure 1 in the revised version of the manuscript. Increased number of mice per group in experiment shown in Figure 1f now allowed to reach significance for fasting glucose levels, which corresponds to increased senescence in aged mice reduced after treatment with MitoTam. Moreover, increased level of leptin (Figure 1g) and insulin (see the answer to Reviewer 2) and their decrease after treatment supports this statement.

We do not conclude that senescent cells and their elimination is the sole mechanism, how MitoTam improves diabetic parameters. However, their elimination contributes to the effect of the agent, since tamoxifen, a known glucose regulator (doi: [10.1016/j.bbrc.2015.07.015](https://doi.org/10.1016/j.bbrc.2015.07.015)), which does not eliminates senescent cells, has only partial effect on diabetic parameters when compared to its mitochondria-targeted counterpart.

We agree with the Reviewer, that detection of senescence in adipose tissue of aged mice would help to assign causality with our prediabetic model. We have therefore measured level of senescence in visceral adipose tissue (VAT). Similarly as for lungs and liver, MitoTam decreases level of senescent markers also in VAT, as is shown in Figures 1b and Supplementary 1c.

In addition, in relation to Figure 1 the fasting glucose levels and the glucose tolerance changes between young and old mice are still described in a manner that implies there is a significant difference (lines 117-118), when in fact there is not - please correct.

We thank the Reviewer for the point. As was mentioned above, during the revision we performed additional experiments with new cohort of aged mice. These data are now included in Figure 1f in the revised version of the manuscript. The results now show no significance in the difference in fasting glucose levels. Due to the low differences between young and aged mice in the experiment with glucose tolerance, we decided to exclude it from manuscript and rewrite the conclusions for Figure 1 in the revised version of the manuscript.

The authors have stridently rebutted the comments of 2 x reviewers that their mouse models is not a true model of T2DM. The mild increase in fasting glycemia in fat-fed animals is a common observation in mice that are insulin resistant, and is somewhat akin to a state of pre-diabetes. However it is not T2DM, and the authors have no evidence for beta cell dysfunction in their animals (an essential component of T2DM), so the mice should be referred to simply as obese and insulin resistant and not diabetic.

We agree with the Reviewer that in contrast to humans, where diabetes is clearly defined by glucose levels and/or glycated hemoglobin assessed under a standardized condition, there are no generally accepted cut-off values for mice. We completely agree that beta-cell dysfunction is an important part of type 2 diabetes pathogenesis, nevertheless it is not required as a part of diagnosis of type 2 diabetes in humans. In fact, insulin levels are often increased due to insulin resistance in the early stages of type 2 diabetes as compared to non-diabetic subjects.

Taken together, we have changed the term diabetic to obese and pre-diabetic in the revised version of the manuscript, as the mice clearly do not have normal glucose values but no cut-off values for diabetes are defined in mice so that we cannot state that these mice have already, in our model, developed T2DM.

As noted in the previous review, once the data relating to Figure 2 was presented appropriately, it is clear that there is a marked (27%) drop in calorie intake in the HFD mice receiving MitoTam. Calorie restriction is known to induce a raft of favorable metabolic effects, such as improve glucose tolerance, reduce lipid accumulation in non-adipose tissues and lower systemic inflammation. MitoTam treatment induced these same metabolic effects in mice, as well as eliminating senescent cells (in vivo and in vitro) and impairing adipogenesis (in vitro). What is impossible to reconcile is the relative importance of the direct effects of MitoTam (on senescent cells and adipogenesis) vs. the reduction in calorie intake induced by this compound. A pair feeding study would resolve this and should be undertaken.

As suggested, pair-feeding studies have been conducted and the results show that both, food intake reduction and other mechanisms contribute to MitoTam action. The data are shown in Figures 2, 3 and in Supplementary Figure 2 in the revised version of the manuscript.

Further to this point, the abstract completely fails to acknowledge the large decrease in food intake induced by MitoTam, and the fact that this likely mediates some of the favorable metabolic effects of MitoTam. Please correct this.

We agree with the reviewer. Based on our new results, we have modified the Abstract in the revised version of the manuscript to correspond better with our findings.

I fail to see the relevance of the change in UCP1 mRNA in subcutaneous adipose tissue. Firstly, while UCP1 protein is clearly important in thermogenesis, there are a raft of other changes (morphological and molecular) that occur during browning of adipose tissue, and without measurement of some of these parameters it is inappropriate to be so definitive about the browning of adipose tissue in response to MitoTam. Notwithstanding this issue, the other major concerns are 1) the browning of adipose tissue is suggested to favourably impact metabolic homeostasis primarily by increasing energy expenditure. The authors assert (lines 386-387) that there is a role for adipose browning in mediating the effects of MitoTam on weight loss, yet their own indirect calorimetry data shows no change in energy expenditure in response to MitoTam treatment. Please clarify how these findings can be reconciled? 2) calorie restriction is known to induce adipose browning (e.g. Fabbiano et al. Cell Metab 24, 434-446, 2016), and given that there was no change at all in UCP1 expression in the SD mice that received MitoTam, it is highly plausible that any potential change in adipose browning is being induced by the reduction in calorie intake in the HFD group and not directly by the MitoTam.

This is relevant point and we thank reviewer for this comment. We assessed additional markers of browning of adipose tissue (expression of PRDM16 and PGC1 α , H/E of SAT). Since we did not observed any changes between control HFD and MitoTam-treated HFD mice with the absence of increased energy expenditure, we agree with the Reviewer that the changes of UCP-1 expression reflect more reduction of calorie/food intake in the HFD MT group. We have modified accordingly our statement in the revised version of the manuscript.

There are still instances where non-significant changes are being described in a manner that implies significance (e.g. line 192-194). This must be addressed.

As suggested by Reviewer 1, we show all measured markers of senescence for all organs in the revised form of the manuscript. To stay consistent, we now show significant results in main figures and non-significant results in supplementary figures. We have reflected these changes in the revised version of the manuscript.

REVIEWERS' COMMENTS

Reviewer #1 (Remarks to the Author):

I appreciate that the authors have done several additional experiments to respond to my comments. I believe most of my comments have been answered, I recommend publication.

Reviewer #2 (Remarks to the Author):

I appreciate the effort of the authors to address the remaining concerns. The addition of a pair-feeding experiment was very useful for better understanding the mechanisms of action of MitoTAM.

I only have a couple of minor points that still need clarification:

1. In lines 219-220 it is stated: "Mice fed HFD for 15 months showed increased body weight that was reduced by 4-week treatment with MitoTam, whereas tamoxifen treatment showed non-significant effect (Fig. 4a)." This statement is misleading. Actually, the body weight loss induced by tamoxifen was not different from the one induced by MitoTam.

2. Fig. 4h/i show reduced liver TAG accumulation after MT but not TX treatment assessed by histology. Suppl Fig. 3d (not S3e as stated in line 240) also shows liver TAG, here as $\mu\text{mol/g}$, which shows no significant difference between the MT and TX treatments. Was there any difference in liver weight between TX and MT which could explain the discrepancy of these results?

Reviewer #3 (Remarks to the Author):

The authors have appropriately addressed all of the concerns raised in my last review. I have no further comments.